# EDITING MODELS WITH TASK ARITHMETIC

**Gabriel Ilharco**[*1]  **Marco Tulio Ribeiro**[2]  **Mitchell Wortsman**[1]  **Suchin Gururangan**[1]
**Ludwig Schmidt**[1,3]  **Hannaneh Hajishirzi**[1,3]  **Ali Farhadi**[1]
[1]University of Washington  [2]Microsoft Research  [3]Allen Institute for AI

## ABSTRACT

Changing how pre-trained models behave—e.g., improving their performance on a downstream task or mitigating biases learned during pre-training—is a common practice when developing machine learning systems. In this work, we propose a new paradigm for steering the behavior of neural networks, centered around *task vectors*. A task vector specifies a direction in the weight space of a pre-trained model, such that movement in that direction improves performance on the task. We build task vectors by subtracting the weights of a pre-trained model from the weights of the same model after fine-tuning on a task. We show that these task vectors can be modified and combined together through arithmetic operations such as negation and addition, and the behavior of the resulting model is steered accordingly. Negating a task vector decreases performance on the target task, with little change in model behavior on control tasks. Moreover, adding task vectors together can improve performance on multiple tasks at once. Finally, when tasks are linked by an analogy relationship of the form "$A$ is to $B$ as $C$ is to $D$", combining task vectors from three of the tasks can improve performance on the fourth, even when no data from the fourth task is used for training. Overall, our experiments with several models, modalities and tasks show that task arithmetic is a simple, efficient and effective way of editing models.

## 1 INTRODUCTION

Pre-trained models are commonly used as backbones of machine learning systems. In practice, we often want to *edit* models after pre-training,[1] to improve performance on downstream tasks [105; 100; 63; 39], mitigate biases or unwanted behavior [85; 59; 82; 71], align models with human preferences [4; 74; 44; 32], or update models with new information [104; 15; 69; 70].

In this work, we present a new paradigm for editing neural networks based on *task vectors*, which encode the information necessary to do well on a given task. Inspired by recent work on weight interpolation [27; 100; 63; 99; 39; 55; 2; 20], we obtain such vectors by taking the weights of a model fine-tuned on a task and subtracting the corresponding pre-trained weights (Figure 1a).

We show that we can edit a variety of models with *task arithmetic*—performing simple arithmetic operations on task vectors (Figure 1b-d). For example, *negating* a vector can be used to remove undesirable behaviors or unlearn tasks, while *adding* task vectors leads to better multi-task models, or even improves performance on a single task. Finally, when tasks form an *analogy* relationship, task vectors can be combined to improve performance on tasks where data is scarce.

**Forgetting via negation.**   Users can negate task vectors to mitigate undesirable behaviors (e.g., toxic generations), or even to forget specific tasks altogether, like OCR. In Section 3, we negate a task vector from a language model fine-tuned on toxic data [77; 8], reducing the proportion of generations classified as toxic, with little change in fluency. We also negate task vectors for image classification tasks, resulting in substantially lower accuracy on the task we wish to forget with little loss on ImageNet accuracy [16].

---

[*]Correspondence to `gamaga@cs.washington.edu`.
[1]We use the term *editing* to refer to any intervention done to a model done after the pre-training stage.

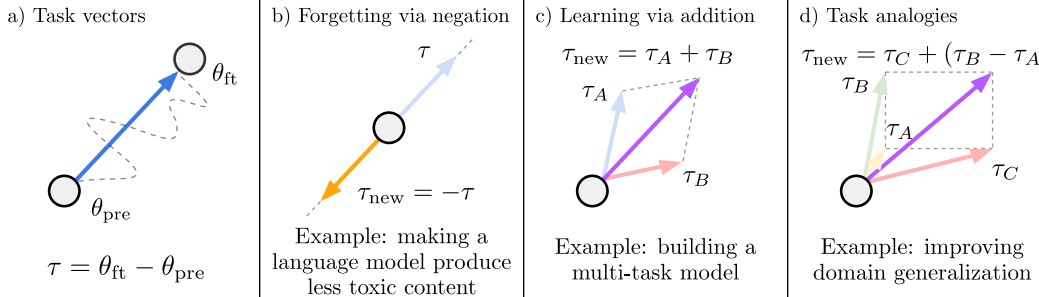

Figure 1: An illustration of task vectors and the arithmetic operations we study for editing models. (a) A task vector is obtained by subtracting the weights of a pre-trained model from the weights of the same model after fine-tuning (Section 2). (b) Negating a task vector degrades performance on the task, without substantial changes in control tasks (Section 3). (c) Adding task vectors together improves the performance of the pre-trained model on the tasks under consideration (Section 4). (d) When tasks form an analogy relationship such as supervised and unsupervised learning on two different data sources, it is possible to improve performance on a supervised target task using only vectors from the remaining three combinations of objectives and datasets (Section 5).

**Learning via addition.** Adding task vectors results in better multi-task models, or improved performance on a single task. In Section 4, we add task vectors from various image models (CLIP, Radford et al. [78]) and compare the performance of the resulting model with using multiple specialized fine-tuned models. We find that the single resulting model can be competitive with using multiple specialized models. Adding two task vectors maintains 98.9% of the accuracy, and the average performance on the entire set of tasks increases as more task vectors are added. Moreover, adding a task vector from a different task can *improve* performance on a target task using text models (T5, Raffel et al. [79]).

**Task analogies.** When we can form task analogies of the form "$A$ is to $B$ as $C$ is to $D$", combining task vectors from the first three tasks improves performance on the fourth, even when little or no training data is available. In Section 5, we show that we can improve domain generalization to a new target task without using labeled data from that task. More specifically, accuracy on a sentiment analysis task improves by combining a task vector from a second sentiment analysis dataset and task vectors produced using unlabeled data from both domains. We also use analogies between classifying pictures and sketches of objects to improve accuracy on subgroups where little or no data is available.

Overall, editing models with task arithmetic is simple, fast and effective. There is no extra cost at inference time in terms of memory or compute, since we only do element-wise operations on model weights. Moreover, vector operations are cheap, allowing users to experiment quickly with multiple task vectors. With task arithmetic, practitioners can reuse or transfer knowledge from models they create, or from the multitude of publicly available models all without requiring access to data or additional training.[2]

## 2 TASK VECTORS

For our purposes, a task is instantiated by a dataset and a loss function used for fine-tuning. Let $\theta_{\text{pre}} \in \mathbb{R}^d$ be the weights of a pre-trained model, and $\theta_{\text{ft}}^t \in \mathbb{R}^d$ the corresponding weights after fine-tuning on task $t$. The task vector $\tau_t \in \mathbb{R}^d$ is given by the element-wise difference between $\theta_{\text{ft}}^t$ and $\theta_{\text{pre}}$, i.e., $\tau_t = \theta_{\text{ft}}^t - \theta_{\text{pre}}$. When the task is clear from context, we omit the identifier $t$, referring to the task vector simply as $\tau$.

Task vectors can be applied to any model parameters $\theta$ from the same architecture, via element-wise addition, with an optional scaling term $\lambda$, such that the resulting model has weights $\theta_{\text{new}} = \theta + \lambda \tau$. In our experiments, the scaling term is determined using held-out validation sets. Note that adding a single task vector to a pre-trained model with $\lambda = 1$ results in the model fine-tuned on that task.

---

[2]Code available at https://github.com/mlfoundations/task_vectors.

Following Ilharco et al. [39], we focus on open-ended models, where it is possible to fine-tune on a downstream task without introducing new parameters (e.g., open-vocabulary image classifiers [78; 42; 76; 3] and text-to-text models [79; 77; 9; 38]). In cases where fine-tuning introduces new parameters (e.g., a new classification head), we could follow Matena & Raffel [63] and merge only the shared weights, but this exploration is left for future work.

**Editing models with task arithmetic.** We focus on three arithmetic expressions over task vectors, as illustrated in Figure 1: negating a task vector, adding task vectors together, and combining task vectors to form analogies. All operations are applied element-wise to the weight vectors.

When *negating* a task vector $\tau$, applying the resulting vector $\tau_{\text{new}} = -\tau$ corresponds to extrapolating between the fine-tuned model and the pre-trained model. The resulting model is worse at the target task, with little change in performance on control tasks (Section 3). *Adding* two or more task vectors $\tau_i$ yields $\tau_{\text{new}} = \sum_i \tau_i$, and results in a multi-task model proficient in all tasks, sometimes even with gains over models fine-tuned on individual tasks (Section 4). Finally, when tasks $A$, $B$, $C$ and $D$ form an analogy in the form "$A$ is to $B$ as $C$ is to $D$", the task vector $\tau_{\text{new}} = \tau_C + (\tau_B - \tau_A)$ improves performance on task $D$, even if there is little or no data for that task (Section 5).

For all operations, the model weights obtained by applying $\tau_{\text{new}}$ are given by $\theta_{\text{new}} = \theta + \lambda\tau_{\text{new}}$, where the scaling term $\lambda$ is determined using held-out validation sets.

## 3 FORGETTING VIA NEGATION

In this section, we show that negating a task vector is an effective way to reduce its performance on a target task, without substantially hurting performance elsewhere. Forgetting or "unlearning" can help mitigate undesired biases learned when pre-training; forgetting tasks altogether may be desirable to comply with regulations or for ethical reasons like preventing an image classifier to recognize faces, or to "read" personal information via OCR.

These interventions should not have a substantial effect on how models behave when processing data outside the scope of the edit [69; 39]. Accordingly, we measure accuracy on *control tasks*, in addition to evaluating on the target tasks from which the task vector originated. Our experiments showcase the effectiveness of negating task vectors for editing image classification and text generation models.

### 3.1 IMAGE CLASSIFICATION

For image classification, we use CLIP models [78] and task vectors from eight tasks studied by Ilharco et al. [39]; Radford et al. [78], ranging from satellite imagery recognition to classifying traffic signs: Cars [47], DTD [12], EuroSAT [36], GTSRB [87], MNIST [51], RESISC45 [10], SUN397 [101], and SVHN [72]. We explore additional tasks including OCR and person identification in Appendix B. For the control task, we use ImageNet [16]. We generate task vectors by fine-tuning on each of the target tasks, as detailed in Appendix B.1.

We compare against two additional baselines, fine-tuning by moving in the direction of increasing loss (i.e., with gradient ascent), as in Golatkar et al. [34]; Tarun et al. [90], and against using a random vector where each layer has the same magnitude as the corresponding layer of task vector. Additional details are in Appendix B.2.

As shown in Table 1, negating the task vectors is the most effective editing strategy for decreasing accuracy on the target task with little impact on the control task. For example, negative task vectors decrease the average target accuracy of ViT-L/14 by 45.8 percentage points with little change in accuracy on the control task. In contrast, using a random vector does not have much impact on target accuracy, while fine-tuning with gradient ascent severely deteriorates performance on control tasks. We present additional results in Appendix B.

### 3.2 TEXT GENERATION

We study whether we can mitigate a particular model behavior by negating a task vector *trained to do that behavior*. In particular, we aim to reduce the amount of toxic generations produced by GPT-2 models of various sizes [77]. We generate task vectors by fine-tuning on data from Civil

Table 1: **Forgetting image classification tasks via negation**. Results are shown for CLIP models, reporting average accuracy (%) on the eight target tasks we wish to forget (Cars, DTD, EuroSAT, GTSRB, MNIST, RESISC45, SUN397 and SVHN), and the control task (ImageNet). Negating task vectors reduce the accuracy of a pre-trained ViT-L/14 by 45.8 percentage points on the target tasks, with little loss on the control task. Additional details and results are shown in Appendix B.

| Method | ViT-B/32 | | ViT-B/16 | | ViT-L/14 | |
|---|---|---|---|---|---|---|
| | Target ($\downarrow$) | Control ($\uparrow$) | Target ($\downarrow$) | Control ($\uparrow$) | Target ($\downarrow$) | Control ($\uparrow$) |
| Pre-trained | 48.3 | 63.4 | 55.2 | 68.3 | 64.8 | 75.5 |
| Fine-tuned | 90.2 | 48.2 | 92.5 | 58.3 | 94.0 | 72.6 |
| Gradient ascent | 2.73 | 0.25 | 1.93 | 0.68 | 3.93 | 16.3 |
| Random vector | 45.7 | 61.5 | 53.1 | 66.0 | 60.9 | 72.9 |
| Negative task vector | 24.0 | 60.9 | 21.3 | 65.4 | 19.0 | 72.9 |

Table 2: **Making language models less toxic with negative task vectors.** Results are shown for the GPT-2 Large model. Negative task vectors decrease the amount of toxic generations by 6×, while resulting in a model with comparable perplexity on a control task (WikiText-103). Additional details and results are shown in Appendix C.

| Method | % toxic generations ($\downarrow$) | Avg. toxicity score ($\downarrow$) | WikiText-103 perplexity ($\downarrow$) |
|---|---|---|---|
| Pre-trained | 4.8 | 0.06 | 16.4 |
| Fine-tuned | 57 | 0.56 | 16.6 |
| Gradient ascent | 0.0 | 0.45 | $>10^{10}$ |
| Fine-tuned on non-toxic | 1.8 | 0.03 | 17.2 |
| Random vector | 4.8 | 0.06 | 16.4 |
| Negative task vector | 0.8 | 0.01 | 16.9 |

Comments [8] where the toxicity score is *higher* than 0.8, and then negating such task vectors. As in Section 3.1, we also compare against baselines that use gradient ascent when fine-tuning [34; 90], and using a random task vector of the same magnitude. Additionally, we compare against fine-tuning on non-toxic samples from Civil Comments (toxicity scores smaller than 0.2), similar to Liu et al. [57]. We measure the toxicity of one thousand model generations with Detoxify [35]. For the control task, we measure the perplexity of the language models on WikiText-103 [66].

As shown in Table 2, editing with negative task vectors is effective, reducing the amount of generations classified as toxic from 4.8% to 0.8%, while maintaining perplexity on the control task within 0.5 points of the pre-trained model. In contrast, fine-tuning with gradient ascent lowers toxic generations by degrading performance on the control task to an unacceptable level, while fine-tuning on non-toxic data is worse than task vectors both in reducing task generations and on the control task. As an experimental control, adding a random vector has little impact either on toxic generations or perplexity on WikiText-103. We present additional experimental details and results in Appendix C.

## 4 LEARNING VIA ADDITION

We now turn our attention to *adding* task vectors, either to build multi-task models that are proficient on multiple tasks simultaneously, or to improve single-task performance. This operation allows us to reuse and transfer knowledge either from in-house models, or from the multitude of publicly available fine-tuned models, without additional training or access to training data. We explore addition on various image classification and natural language processing tasks.

### 4.1 IMAGE CLASSIFICATION

We start with the same eight models used in Section 3, fine-tuned on a diverse set of image classification tasks (Cars, DTD, EuroSAT, GTSRB, MNIST, RESISC45, SUN397 and SVHN). In Figure 2, we

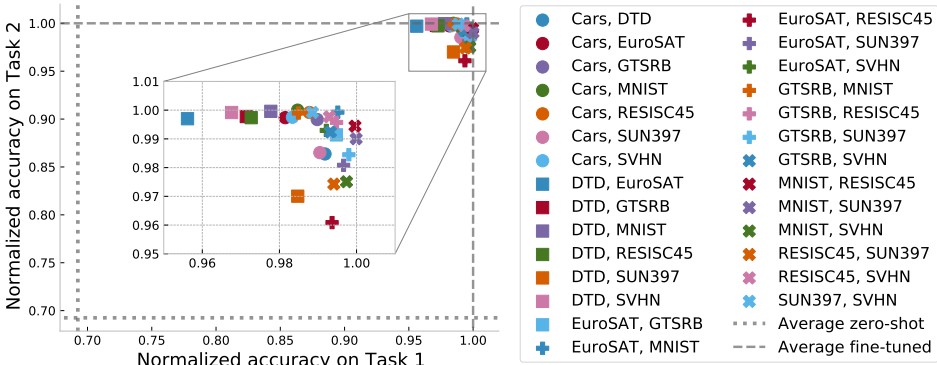

Figure 2: **Adding pairs of task vectors** from image classification tasks. Adding task vectors from two tasks improves accuracy on both, resulting in a single model that is competitive with using two specialized fine-tuned models.

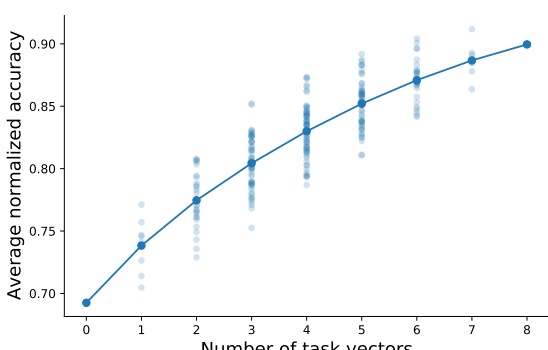

Figure 3: **Adding task vectors builds multi-task models** for image classification tasks. Accuracy is averaged over all downstream tasks. When more task vectors are available, better multi-task vectors can be built. Each point represents an experiment with a subset of the eight tasks we study, and the solid line connects the average performance for each subset size. Recall that the average normalized accuracy of using multiple fine-tuned models is always one. Additional details and experiments are in Appendix D.

show the accuracy obtained by adding all pairs of task vectors from these tasks. To account for the difference in difficulty of the tasks, we normalize accuracy on each task by the accuracy of the model fine-tuned on that task. After normalizing, the performance of fine-tuned models on their respective tasks is one, and so the average performance of using multiple specialized models is also one. As shown in Figure 2, adding pairs of task vectors leads to a single model that outperforms the zero-shot model by a large margin, and is competitive with using two specialized models (98.9% normalized accuracy on average).

Beyond pairs of tasks, we explore adding task vectors for *all* possible subsets of the tasks ($2^8$ in total). In Figure 3, we show how the normalized accuracy of the resulting models, averaged over all the eight tasks. As the number of available task vectors increases, better multi-task models can be produced. When all task vectors are available, the best model produced by adding task vectors reaches an average performance of 91.2%, despite compressing several models into one. Additional experiments and details are presented in Appendix D.

## 4.2 NATURAL LANGUAGE PROCESSING

In addition to building multi-task models, we explore whether adding task vectors is a useful way of improving performance on a single target task. Towards this goal, we first fine-tune T5-base models on four tasks from the GLUE benchmark [93], as in Wortsman et al. [99]. Then, we search for compatible checkpoints on Hugging Face Hub, finding 427 candidates in total. We try adding each of the corresponding task vectors to our fine-tuned models, choosing the best checkpoint and scaling coefficient based on held-out validation data. As shown in Table 3, adding task vectors can *improve* performance on target tasks, compared to fine-tuning. Additional details and experiments—including building multi-task models from public checkpoints from Hugging Face Hub—are presented in Appendix D.

Table 3: **Improving performance on target tasks with external task vectors.** For four text classification tasks from the GLUE benchmark, adding task vectors downloaded from the Hugging Face Hub can improve accuracy of fine-tuned T5 models. Appendix D.6 shows additional details.

| Method | MRPC | RTE | CoLA | SST-2 | Average |
|---|---|---|---|---|---|
| Zero-shot | 74.8 | 52.7 | 8.29 | 92.7 | 57.1 |
| Fine-tuned | 88.5 | 77.3 | 52.3 | 94.5 | 78.1 |
| Fine-tuned + task vectors | 89.3 (+0.8) | 77.5 (+0.2) | 53.0 (+0.7) | 94.7 (+0.2) | 78.6 (+0.5) |

Table 4: **Improving domain generalization with task analogies.** Using an auxiliary task for which labeled data is available and unlabeled data from both the auxiliary and the target datasets, task analogies improve the accuracy for multiple T5 models and two sentiment analysis target tasks [102; 65], without using any labeled data from the target tasks.

| Method | target = Yelp | | | target = Amazon | | |
|---|---|---|---|---|---|---|
| | T5-small | T5-base | T5-large | T5-small | T5-base | T5-large |
| Fine-tuned on auxiliary | 88.6 | 92.3 | 95.0 | 87.9 | 90.8 | 94.8 |
| Task analogies | 89.9 | 93.0 | 95.1 | 89.0 | 92.7 | 95.2 |
| Fine-tuned on target | 91.1 | 93.4 | 95.5 | 90.2 | 93.2 | 95.5 |

## 5 TASK ANALOGIES

In this section, we explore task analogies in the form "$A$ is to $B$ as $C$ is to $D$", and show that task arithmetic using vectors from the first three tasks improves performance on task $D$ even if little or not data for that task is available.

**Domain generalization.** For many target tasks, gathering unlabeled data is easier and cheaper than collecting human annotations. When labeled data for a *target* task is not available, we can use task analogies to improve accuracy on the target task, using an *auxiliary* task for which there is labeled data and an unsupervised learning objective. For example, consider the target task of sentiment analysis using data from Yelp [102]. Using task analogies, we can construct a task vector $\hat{\tau}_{\text{yelp; sent}} = \tau_{\text{amazon; sent}} + (\tau_{\text{yelp; lm}} - \tau_{\text{amazon; lm}})$, where $\tau_{\text{amazon; sent}}$ is obtained by fine-tuning on labeled data from an auxiliary task (sentiment analysis using data from Amazon; McAuley & Leskovec [65]), and $\tau_{\text{yelp; lm}}$ and $\tau_{\text{amazon; lm}}$ are task vectors obtained via (unsupervised) language modeling on the inputs from both datasets.

In Table 4, we show that using such task analogies improves accuracy of T5 models at multiple scales, both for Amazon and Yelp binary sentiment analysis as target tasks. We empirically found that giving a higher weight to the sentiment analysis task vector led to higher accuracy, and we thus used two independent scaling coefficients for these experiments—one for the sentiment analysis task vector and one for both the language modeling task vectors. More details are presented in Appendix E.1. Using task vectors outperforms fine-tuning on the remaining auxiliary sentiment analysis task for all models and datasets, approaching the performance of fine-tuning on the target task.

**Subpopulations with little data.** There is often some inherent scarcity in certain data subpopulations—for example, images of lions in indoor settings are more rare, compared to lions in outdoor settings or dogs in general (indoor or outdoors). Whenever such subpopulations admit analogies to others with more abundant data (as in this case), we can apply task analogies, e.g., $\hat{\tau}_{\text{lion indoors}} = \tau_{\text{lion outdoors}} + (\tau_{\text{dog indoors}} - \tau_{\text{dog outdoor}})$.

We explore this scenario by creating four subpopulations, using 125 overlapping classes between ImageNet and a dataset of human sketches [22]. We split these classes in two subsets of roughly equal size, creating four subpopulations $A$, $B$, $C$ and $D$, where the pairs $(A, C)$ and $(B, D)$ share the same classes, and $(A, B)$ and $(C, D)$ share the same style (photo-realistic images or sketches). Although these subpopulations have many classes in our experiments, we use the simplified subsets

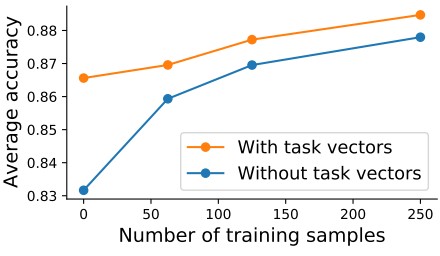

Figure 4: **Learning about subpopulations via analogy**. Combining task vectors from related subpopulations improves accuracy on the target subpopulation, when little or no data from the target supopulation is available. Accuracy is averaged over the four target subpopulations and three CLIP models. Additional details are in Appendix E.3.

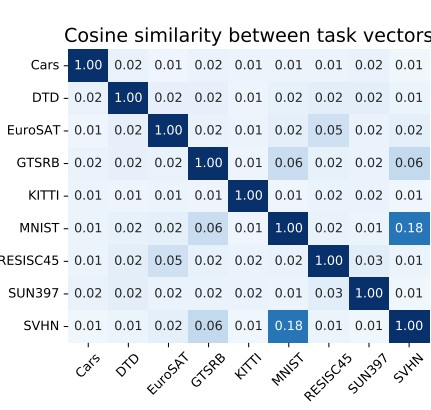

Figure 5: **Task vectors are typically close to orthogonal.** The plot shows the cosine similarities between vectors for different tasks, using CLIP. The largest deviations from orthogonality are found when tasks are similar to each other, for instance, for MNIST, SVHN and GTSRB—where recognizing digits is either the task itself (MNIST and SVHN), or a capability needed to solve the task (GTSRB, where the task is traffic sign recognition)—and EuroSAT and RESISC45, two satellite imagery recognition datasets.

"real dog", "real lion", "sketch dog" and "sketch lion" as a running example. We present more details and samples in Appendix E.3.

Given a target subpopulation, we create task vectors by fine-tuning three models independently on the remaining subpopulations, and then combine them via task arithmetic, e.g., $\hat{\tau}_{\text{sketch lion}} = \tau_{\text{sketch dog}} + (\tau_{\text{real lion}} - \tau_{\text{real dog}})$ for the target subpopulation "sketch lion". We show the results in Figure 4, averaged over the four target subpopulations. Compared to the pre-trained model, task vectors improve accuracy by 3.4 percentage points on average. Moreover, when some data from the target subpopulation is available for fine-tuning, starting from the edited model leads to consistently higher accuracy than starting from the pre-trained model. The gains from analogies alone (with no additional data) are roughly the same as that of collecting and annotating around one hundred training samples for the target subpopulation.

**Kings and queens.** We explore whether an image classifier can learn a new categories (e.g., "king") using data from three related classes that form an analogy relationship (e.g., "queen", "man" and "woman"). Our results are presented in Appendix E.2, showing that task analogies yield large gains in accuracy over pre-trained models on the new target category, despite having no training data for it.

## 6 DISCUSSION

In this section, we provide further insight into previous results by exploring the similarity between task vectors for different tasks, as well as the impact of different learning rates and random seeds. Additional analysis are presented in Appendix A, including discussions on the connection between ensembles and weight averaging. We conclude by discussing some limitations of our approach.

**Similarity between task vectors.** In Figure 5, we explore the cosine similarity between task vectors for different tasks, in an effort to understand how multiple models can be collapsed into a single multi-task model via addition (Section 4). We observe that vectors from different tasks are typically close to orthogonal, and speculate that this enables the combination of task vectors via addition with minimal interference. We also observe higher cosine similarities when tasks are semantically similar to each other. For example, the largest cosine similarities in Figure 5 (left) are between MNIST, SVHN and GTSRB, where recognizing digits is essential for the tasks, and between EuroSAT and RESISC45, which are both satellite imagery recognition datasets. This similarity in "task space" could help explain some results in Ilharco et al. [39], where interpolating the weights of a model

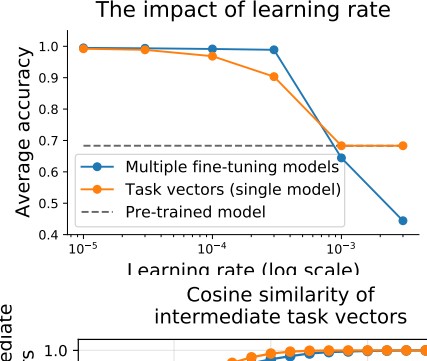

Figure 6: **The impact of learning rate when fine-tuning.** When adding task vectors from CLIP ViT-L/14 models fine-tuned on MNIST and EuroSAT, lower learning rates make the best use of the fine-tuned models, and also correspond to the highest accuracies of the fine-tuned models on the target task.

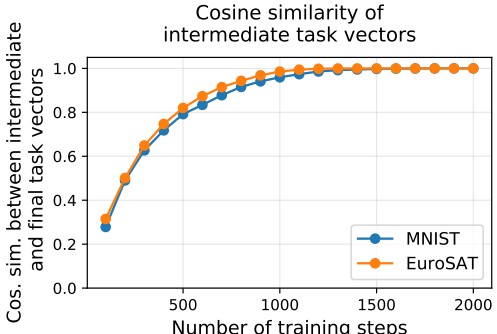
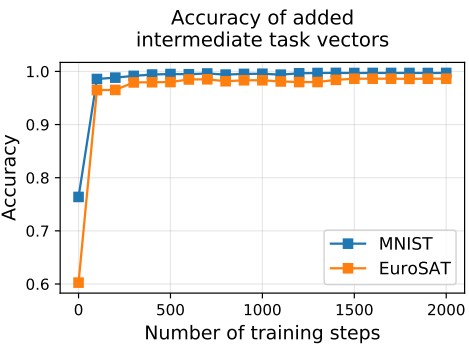

Figure 7: **How task vectors evolve throughout fine-tuning.** Left: the cosine similarity between the final task vector and task vectors produced at intermediate points during fine-tuning. Right: Accuracy obtained by adding intermediate task vectors from MNIST and EuroSAT. Adding intermediate task vectors can lead to high accuracy, despite fine-tuning for substantially fewer steps.

fine-tuned on one task and the pre-trained model weights—in our terminology, applying a single task vector—sometimes improves accuracy on a different task for which no data is available (e.g., applying the MNIST task vector improves accuracy on SVHN).

**The impact of the learning rate.** In Figure 6, we observe that increasing the learning rate degrades accuracy both when using task vectors and when fine-tuning individual models, but the decrease is more gradual for individual models. These findings align with those of [99], who observed that accuracy decreases on the linear path between two fine-tuned models when using a larger learning rate. Thus, while larger learning rates may be acceptable when fine-tuning individual models, we recommend more caution when using task vectors. Further, we hypothesize that larger learning rates may explain some of the variance when adding vectors from natural language processing tasks, where we take models fine-tuned by others in the community.

**The evolution of task vectors throughout fine-tuning.** In Figure 7, we show how task vectors evolve throughout fine-tuning. Intermediate task vectors converge rapidly to the direction of the final task vector obtained at the end of fine-tuning. Moreover, the accuracy of the model obtained by adding intermediate task vectors from two image classification tasks saturates after just a few hundred steps. These results suggest that using intermediate task vectors can be a useful way of saving compute with little harm in accuracy.

**Limitations.** Task vectors are restricted to models with the same architecture, since they depend on element-wise operations on model weights. Further, in all of our experiments we perform arithmetic operations only on models fine-tuned from the same pre-trained initialization, although emerging work shows promise in relaxing this assumption [2]. We also note that some architectures are very popular, and have "standard" initializations—e.g., at the time of writing there are over 3,000 models on Hugging Face Hub fine-tuned from the same BERT-base initialization [17], and over 800 models fine-tuned from the same T5-small initialization.

# 7 RELATED WORK

**The loss landscape and interpolating weights.** The geometry of neural network loss surfaces has attracted the interest of several authors in recent years [54; 28; 21; 48; 25; 13; 98; 7; 23; 55; 60].

Despite neural networks being non-linear, previous work has empirically found that interpolations between the weights of two neural networks can maintain their high accuracy, provided these two neural networks share part of their optimization trajectory [27; 40; 73; 26; 99; 11; 39].

In the context of fine-tuning, accuracy increases steadily when gradually moving the weights of a pre-trained model in the direction of its fine-tuned counterpart [100; 63; 39]. Beyond a single task, Matena & Raffel [63]; Ilharco et al. [39] found that when multiple models are fine-tuned on different tasks from the same initialization, averaging their weights can improve accuracy on the fine-tuning tasks. Similar results were found by Li et al. [55] when averaging the parameters of language models fine-tuned on various domains. Choshen et al. [11] showed that "fusing" fine-tuned models by averaging their weights creates a better starting point for fine-tuning on a new downstream task. Wortsman et al. [99] found that averaging the weights of models fine-tuned on multiple tasks can increase accuracy on a new downstream task, without any further training. These findings are aligned with results shown in Section 4. In this work, we go beyond interpolating between models, examining extrapolating between models and additional ways of combining them (Sections 3 and 5).

**Model interventions.** Considering that re-training models is prohibitively expensive in most circumstances, several authors have studied more efficient methods for modifying a model's behavior with interventions after pre-training, referring to this process by different names, such as patching [33; 89; 39; 71], editing [85; 69; 70], aligning [74; 4; 44; 32], or debugging [82; 30]. In contrast to previous literature, our work provides a unique way of editing models, where capabilities can be added or deleted in a modular and efficient manner by re-using fine-tuned models. Closer to our work is that of Subramani et al. [88], who explore steering language models with vectors added to its hidden states. In contrast, our work applies vectors in the weight space of pre-trained models and does not modify the standard fine-tuning procedure.

**Task embeddings.** Achille et al. [1]; Vu et al. [91; 92], inter alia, explored strategies for representing tasks with continuous embeddings, in order to to predict task similarities and transferability, or to create taxonomic relations. While the task vectors we build could be used for such purposes, our main goal is to use them as tools for steering the behavior of pre-trained models. Additionally, Lampinen & McClelland [50] propose a framework for adapting models based on relationships between tasks. In contrast to their work, our framework uses only linear combinations of model weights.

## 8 CONCLUSION

In this paper we introduce a new paradigm for editing models based on arithmetic operations over *task vectors*. For various vision and NLP models, *adding* multiple specialized task vectors results in a single model that performs well on all target tasks, or even improves performance on a single task. *Negating* task vectors allows users to remove undesirable behaviors, e.g., toxic generations, or even forget specific tasks altogether, while retaining performance everywhere else. Finally, *task analogies* leverage existing data to improve performance on domains or subpopulations where data is scarce.

Arithmetic operations over task vectors only involve adding or subtracting model weights, and thus are efficient to compute, especially when compared to alternatives that involve additional fine-tuning. Thus, users can easily experiment with various model edits, recycling and transferring knowledge from large collections of publicly available fine-tuned models. Since these operations result in a single model of the same size, they incur no extra inference cost. Our code is available at https://github.com/mlfoundations/task_vectors.

## ACKNOWLEDGEMENTS

We thank Alex Fang, Ari Holtzman, Colin Raffel, Dhruba Ghosh, Jesse Dodge, Margaret Li, Ofir Press, Sam Ainsworth, Sarah Pratt, Stephen Mussmann, Tim Dettmers, and Vivek Ramanujan for helpful discussion and comments on the paper. This work is in part supported by the NSF AI Institute for Foundations of Machine Learning (IFML), Open Philanthropy, NSF IIS 1652052, NSF IIS 17303166, NSF IIS 2044660, ONR N00014-18-1-2826, ONR MURI N00014- 18-1-2670, DARPA N66001-19-2-4031, DARPA W911NF-15-1-0543, the Sloan Fellowship and gifts from AI2.

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

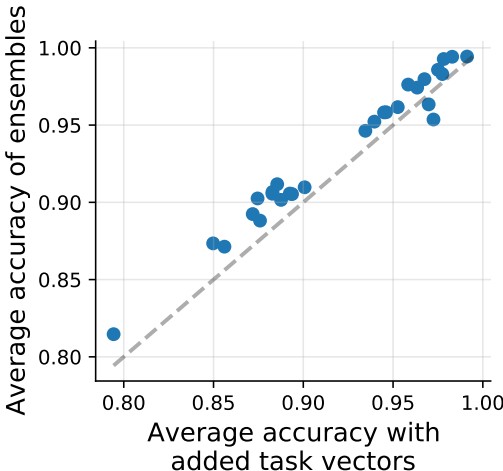

Figure 8: When adding two task vectors, the performance of the resulting model approximates the performing of ensembling the corresponding fine-tuned models.

## A  THE LOSS LANDSCAPE, WEIGHT AVERAGING AND ENSEMBLES

When two neural networks share part of their optimization trajectory—such as when fine-tuning from the same pre-trained initialization—previous work found that performance does not decrease substantially when linearly interpolating between their weights [27; 40; 73; 26; 99; 11; 39]. Applying a task vector—and any vectors produced via the arithmetic expressions we study in this work—is equivalent to a linear combination of the pre-trained model and the fine-tuned models used to generate the task vectors, since only linear operations are used. Interpolating between the weights of a fine-tuned model and its pre-trained counterpart as in Wortsman et al. [100]; Ilharco et al. [39] is equivalent to applying a single task vector, and adding different task vectors is equivalent to a weighted average of all models, similar to experiments from Wortsman et al. [99]; Ilharco et al. [39]; Li et al. [55]. Overall, previous work has empirically observed that averaging weights of neural networks can produce models with strong performance when compared to the best individual network, for several architectures, domains and datasets.

Our motivation for studying task vectors is also well aligned with findings of Lucas et al. [61]; Ilharco et al. [39], who observed that performance steadily increases on the linear path between a model before and after training.[3] This indicates that the direction from the pre-trained to the fine-tuned model is such that movement in that direction directly translates to performance gains on the fine-tuning task. Moreover, Ilharco et al. [39] found that linear interpolations between a pre-trained model and a fine-tuned model are able to preserve accuracy on tasks that are unrelated to fine-tuning, while greatly improving accuracy on the fine-tuning task compared to the pre-trained model. That accuracy on the fine-tuning task and on unrelated tasks are independent of each other along the linear path between pre-trained and fine-tuned models is well aligned with our results on from Section 3, where we find that *extrapolating* from the pre-trained model away from the fine-tuned model leads to worse performance on the fine-tuning task with little change in behavior on control tasks.

Finally, we highlight the connection between linear combinations of neural network weights and the well-established practice of *ensembling* their predictions.[4] This connection is discussed in depth by Wortsman et al. [100; 99], and we briefly revisit it in the context of adding task vectors. First, recall that the arithmetic operations we study result in linear combinations of model weights. As shown by Wortsman et al. [100], in certain regimes, the result from linearly combining the weights of neural network approximate ensembling their outputs. This approximation holds whenever the

---

[3]This property of neural networks is sometimes referred to as Monotonic Linear Interpolation (MLI) [61].

[4]For the sake of completion, the ensemble of two models $f$ with weights $\theta_1$ and $\theta_2$ for an input $x$ is given by $(1 - \alpha)f_{\theta_1}(x) + \alpha f_{\theta_2}(x)$, for some mixing coefficient $\alpha$. Ensembling two classification models is typically done by averaging the logits produced by the models.

loss can be locally approximated by a linear expansion, which is referred to as the NTK regime [41]. Moreover, as shown by Fort et al. [26], this linear expansion becomes more accuracy in the later phase of training neural networks, which closely resembles fine-tuning. When the approximation holds exactly, weight averaging and ensembles are exactly equivalent [100]. This connection is further studied analytically and empirically by Wortsman et al. [99].

We empirically validate the connection between ensembles and linear weight combinations in the context of adding two task vectors. Note that the model resulting from adding two task vectors with a scaling coefficient $\lambda = 0.5$ is equivalent to a uniform average of the weights of the fine-tuned models.[5] We then investigate whether accuracy of the model obtained using the task vectors correlates with the accuracy of ensembling the fine-tuned models, as predicted by theory. As shown in Figure 8, we indeed observe that the accuracy of the model produced by adding two task vectors closely follows the accuracy of the corresponding ensemble. We observe a slight bias towards higher accuracy for the ensembles on average, and that the two quantities are also strongly correlated, with a Pearson correlation of 0.99.

## B    FORGETTING IMAGE CLASSIFICATION TASKS

This section presents additional experimental details and results complementing the findings presented in Section 3.1, showcasing the effect of negating task vectors from image classification tasks.

### B.1    EXPERIMENTAL DETAILS

We follow the same procedure from [39] when fine-tune CLIP models [78]. Namely, we fine-tune for 2000 iterations with a batch size of 128, learning rate 1e-5 and a cosine annealing learning rate schedule with 200 warm-up steps and the AdamW optimizer [58; 75], with weight decay 0.1. When fine-tuning, we freeze the weights of the classification layer output by CLIP's text encoder, so that we do not introduce additional learnable parameters, as in [39]. As shown by [39], freezing the classification layer does not harm accuracy. After fine-tuning, we evaluate scaling coefficients $\lambda \in \{0.0, 0.05, 0.1, \cdots, 1.0\}$, choosing the highest value such that the resulting model still retains at least 95% of the accuracy of the pre-trained model on the control task.

### B.2    BASELINES

We contrast our results with two baselines, fine-tuning with gradient ascent as in Golatkar et al. [34]; Tarun et al. [90], and against using a random vector of the same magnitude as the task vector on a layer-by-layer basis.

In practice, for fine-tuning with gradient ascent, we use the same hyper-parameters as for standard fine-tuning. However, instead of optimizing to minimize the cross-entropy loss $\ell = \mathbb{E}_{x,y \in \mathcal{D}}[-\log f(x)_y]$, we optimize to minimize its negative value, $\ell_{\text{neg}} = -\ell = \mathbb{E}_{x,y \in \mathcal{D}}[\log f(x)_y]$, where $x, y$ are samples in the dataset $\mathcal{D}$ and $f(x)_y$ is the probability assigned by the model $f$ that the inputs $x$ belong to label $y$. This is equivalent to performing gradient ascent on $\ell$.

For the random vector baseline, we first compute the different between the parameters of the pre-trained and fine-tuned models for each layer $L$, $\tau^{(L)} = \theta_{\text{ft}}^{(L)} - \theta_{\text{pre}}^{(L)}$. Then, we draw a new vector $\tau_{\text{rand}}^{(L)} \sim \mathcal{N}(0, I)$ where each element is drawn from a normal distribution with mean 0 and variance 1. We then scale this vector so it has the same magnitude as $\tau^{(L)}$, resulting in $\tau_{\text{scaled}}^{(L)} = \tau_{\text{rand}}^{(L)} \frac{||\tau^{(L)}||}{||\tau_{\text{rand}}^{(L)}||}$.

Finally, we concatenate all the vectors $\tau_{\text{scaled}}^{(L)}$ for all layers to form a new vector withe the same dimensionality as the model parameters $\theta$, which is used in the same way as task vectors.

### B.3    BREAKDOWN PER TASK

Tables 5, 6 and 7 show a breakdown of accuracy for the eight tasks and the three CLIP models we examine.

---

[5] $\theta_{\text{pre}} + 0.5(\tau_1 + \tau_2) = \theta_{\text{pre}} + 0.5((\theta_1 - \theta_{\text{pre}}) + (\theta_2 - \theta_{\text{pre}})) = 0.5(\theta_1 + \theta_2)$.

Table 5: Forgetting via negation on image classification tasks. Results are shown for a CLIP ViT-L/14 model [78], reporting accuracy on both the target (T) and control (C) tasks.

| Method | Cars | | DTD | | EuroSAT | | GTSRB | | MNIST | | RESISC45 | | SUN397 | | SVHN | |
|---|---|---|---|---|---|---|---|---|---|---|---|---|---|---|---|---|
| | T↓ | C↑ | T↓ | C↑ | T↓ | C↑ | T↓ | C↑ | T↓ | C↑ | T↓ | C↑ | T↓ | C↑ | T↓ | C↑ |
| Pre-trained | 77.8 | 75.5 | 55.4 | 75.5 | 60.2 | 75.5 | 50.6 | 75.5 | 76.4 | 75.5 | 71.0 | 75.5 | 68.3 | 75.5 | 58.6 | 75.5 |
| Fine-tuned | 92.8 | 73.1 | 83.7 | 72.3 | 99.2 | 70.5 | 99.3 | 73.1 | 99.8 | 72.9 | 96.9 | 73.8 | 82.4 | 72.7 | 98.0 | 72.6 |
| Neg. gradients | 0.00 | 4.82 | 2.13 | 0.10 | 9.26 | 1.07 | 1.19 | 0.07 | 9.80 | 67.0 | 2.14 | 0.07 | 0.25 | 0.00 | 6.70 | 57.2 |
| Random vector | 72.0 | 73.3 | 52.1 | 72.2 | 59.7 | 73.5 | 43.4 | 72.5 | 74.8 | 72.8 | 70.8 | 73.0 | 66.9 | 72.7 | 47.1 | 72.9 |
| Neg. task vector | 32.0 | 72.4 | 26.7 | 72.2 | 7.33 | 73.3 | 6.45 | 72.2 | 2.69 | 74.9 | 19.7 | 72.9 | 50.8 | 72.6 | 6.71 | 72.7 |

Table 6: Forgetting via negation on image classification tasks. Results are shown for a CLIP ViT-B/16 model [78], reporting accuracy on both the target (T) and control (C) tasks.

| Method | Cars | | DTD | | EuroSAT | | GTSRB | | MNIST | | RESISC45 | | SUN397 | | SVHN | |
|---|---|---|---|---|---|---|---|---|---|---|---|---|---|---|---|---|
| | T↓ | C↑ | T↓ | C↑ | T↓ | C↑ | T↓ | C↑ | T↓ | C↑ | T↓ | C↑ | T↓ | C↑ | T↓ | C↑ |
| Pre-trained | 64.6 | 68.3 | 44.9 | 68.3 | 53.9 | 68.3 | 43.4 | 68.3 | 51.6 | 68.3 | 65.8 | 68.3 | 65.5 | 68.3 | 52.0 | 68.3 |
| Fine-tuned | 87.0 | 61.9 | 82.3 | 57.5 | 99.1 | 56.0 | 99.0 | 54.7 | 99.7 | 55.2 | 96.4 | 62.2 | 79.0 | 61.7 | 97.7 | 56.8 |
| Neg. gradients | 0.36 | 0.11 | 2.13 | 0.09 | 9.26 | 0.14 | 0.71 | 0.10 | 0.04 | 1.20 | 2.60 | 0.10 | 0.25 | 0.00 | 0.08 | 3.69 |
| Rand. task vector | 61.0 | 65.6 | 43.9 | 66.3 | 51.7 | 66.2 | 43.1 | 65.0 | 51.6 | 68.3 | 63.6 | 65.6 | 63.7 | 65.2 | 46.2 | 65.5 |
| Neg. task vector | 30.8 | 65.4 | 26.5 | 65.6 | 12.3 | 65.8 | 9.53 | 65.8 | 9.55 | 65.4 | 26.5 | 65.1 | 48.6 | 65.1 | 6.43 | 65.4 |

We observe qualitatively similar results in all cases. Similarly to what is observed in [39], we also see that results improve with scale: on average, the largest model, ViT-L/14, achieves *lower* accuracy on the target tasks, compared to the smaller models.

## B.4 Additional Visualizations

In Figure 9, we show how accuracy on the target and control tasks vary as we change the scaling coefficients $\lambda$, both for the task vector obtained by fine-tuning on the target task and for a random vector of the same magnitude.

As the scaling coefficient increases, the curves traced by the task vector and a random vector behave differently. For task vectors, performance on the target tasks ($y$-axis) initially decreases faster than performance on the control task ($x$-axis), so there exists models with high accuracy on the control task but low accuracy on the target task. In contrast, such points do not exist in the curves traced by random vectors, which move more linearly towards the origin. In practice, this means forgetting is effective for task vectors obtained by fine-tuning, but not for random vectors.

Table 7: Forgetting via negation on image classification tasks. Results are shown for a CLIP ViT-B/32 model [78], reporting accuracy on both the target (T) and control (C) tasks.

| Method | Cars | | DTD | | EuroSAT | | GTSRB | | MNIST | | RESISC45 | | SUN397 | | SVHN | |
|---|---|---|---|---|---|---|---|---|---|---|---|---|---|---|---|---|
| | T↓ | C↑ | T↓ | C↑ | T↓ | C↑ | T↓ | C↑ | T↓ | C↑ | T↓ | C↑ | T↓ | C↑ | T↓ | C↑ |
| Pre-trained | 59.6 | 63.4 | 44.1 | 63.4 | 45.9 | 63.4 | 32.5 | 63.4 | 48.7 | 63.4 | 60.7 | 63.4 | 63.2 | 63.4 | 31.5 | 63.4 |
| Fine-tuned | 79.2 | 55.2 | 78.7 | 49.3 | 98.6 | 47.2 | 98.5 | 39.1 | 99.6 | 42.5 | 95.0 | 53.2 | 75.1 | 54.6 | 97.2 | 44.7 |
| Neg. gradients | 0.01 | 0.11 | 2.13 | 0.10 | 9.26 | 0.10 | 1.19 | 0.07 | 0.00 | 1.22 | 2.60 | 0.10 | 0.25 | 0.01 | 6.38 | 0.29 |
| Rand. task vector | 54.1 | 60.9 | 39.9 | 61.5 | 45.8 | 63.4 | 27.9 | 60.7 | 48.3 | 63.4 | 57.1 | 60.9 | 61.3 | 60.5 | 31.2 | 60.7 |
| Neg. task vector | 36.0 | 61.1 | 27.8 | 60.2 | 13.6 | 61.3 | 8.13 | 61.4 | 16.7 | 60.7 | 31.7 | 61.0 | 50.7 | 60.5 | 7.65 | 61.0 |

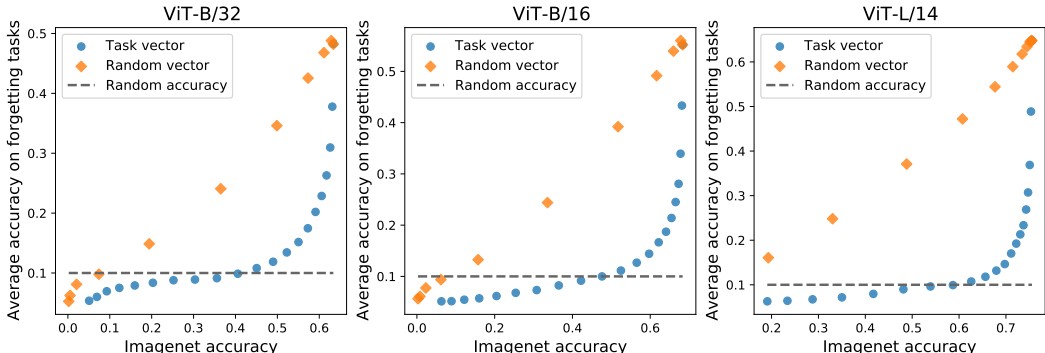

Figure 9: Comparison between task vectors and random vectors for forgetting image classification tasks.

Table 8: The effect of semantic overlap with the control task in forgetting experiments on image classification tasks. Results are shown for a CLIP ViT-L/14 model, reporting accuracy both on the target task and control task (Ctrl, ImageNet).

| Method | Without filtering | | | | With filtering | | | |
|---|---|---|---|---|---|---|---|---|
| | Cars (↓) | Ctrl (↑) | SUN397 (↓) | Ctrl (↑) | Cars (↓) | Ctrl (↑) | SUN397 (↓) | Ctrl (↑) |
| Pre-trained | 77.8 | 75.5 | 68.3 | 75.5 | 77.8 | 75.5 | 68.3 | 76.1 |
| Fine-tuned | 92.8 | 73.1 | 82.4 | 72.7 | 92.8 | 73.3 | 82.4 | 73.1 |
| Neg. task vector | 32.0 | 72.4 | 50.8 | 72.6 | 32.0 | 72.5 | 48.1 | 72.4 |

## B.5 THE EFFECT OF CLASS OVERLAP

In Tables 5, 7, 6, we observe that the tasks where forgetting via task vectors is least effective are tasks where the distribution of images is closer to ImageNet, SUN397 [101], a scene understanding dataset with classes such as "church" and "tower", and Stanford Cars [47], a dataset with with many car categories such as "2012 Tesla Model S" or "2012 BMW M3 coupe". One reasonable hypothesis is that forgetting is less effective for those tasks due to the overlap with the images from the control tasks.

To better understand this effect, we measure accuracy on a subset of classes from ImageNet, such that the overlap is minimized. Concretely, we exclude nodes from the WordNet hierarchy from which the ImageNet classes are based.[6] For the Cars dataset, we exclude the all subnodes under the node "wheeled vehicle" (e.g., "minivan", "jeep", "limousine"). For SUN397, we exclude all subnodes under the nodes "structure" and "geological formation". As shown in Table 8, we do not observe large differences after filtering.

## B.6 INTERPOLATING WITH A MODEL FINE-TUNED WITH GRADIENT ASCENT

One baseline explored in the experiments is fine-tuning with gradient ascent, as explored in Golatkar et al. [34]; Tarun et al. [90]. Our results show that this strategy is effective at reducing the accuracy on treatment tasks, but also substantially deteriorates accuracy on the control task, which is undesirable.

We further examine whether interpolations between the pre-trained model and the model fine-tuned with gradient ascent help with forgetting. Our results, shown in Figure 10, indicate that interpolations greatly mitigate the low accuracy on the control task of the fine-tuned model, leading to even better accuracy trade-offs than the solutions obtained by extrapolation with standard fine-tuning.

---

[6]A visualization is available at https://observablehq.com/@mbostock/imagenet-hierarchy

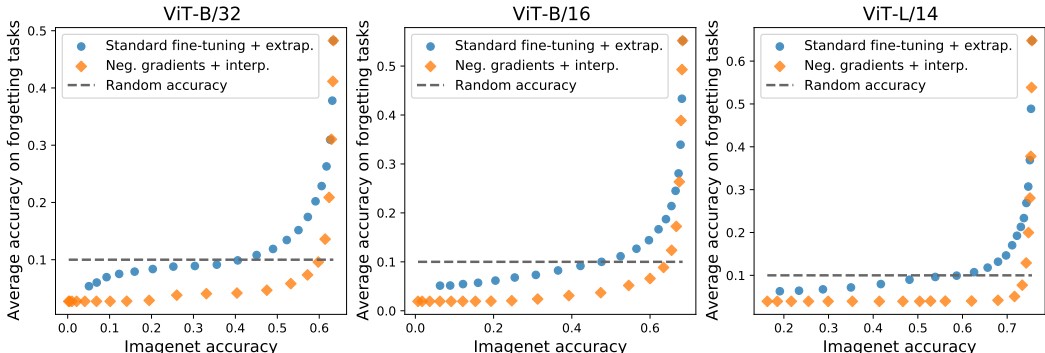

Figure 10: Comparison with interpolations between the pre-trained model and models fine-tuned with gradient ascent.

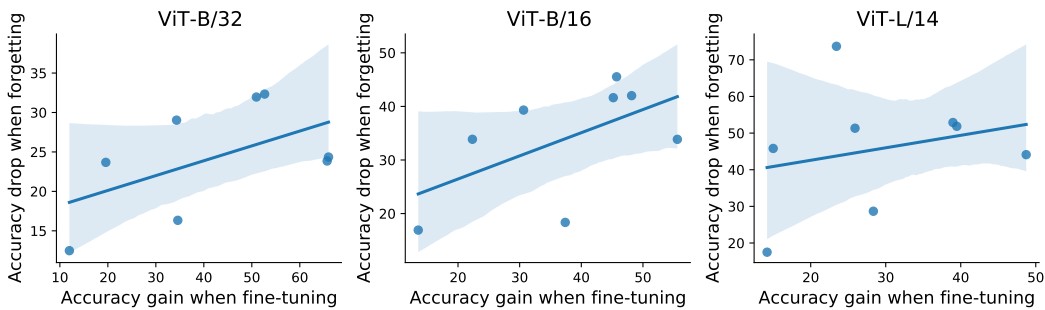

Figure 11: Correlation between the gain in accuracy from fine-tuning and the drop in accuracy when subtracting the corresponding task vector for image classification tasks.

### B.7 WHEN NEGATING TASK VECTORS WORKS BEST

We observe a positive correlation between the gain in accuracy from fine-tuning and the drop in accuracy when subtracting the corresponding task vector, both in comparison with the pre-trained model (Figure 11). We speculate that the reason for this correlation is that when the gains from fine-tuning are small, the task vector provides a less clear direction of improvement, and the opposite direction thus provides a less clear direction of performance deterioration. In the extreme case where fine-tuning does not improve accuracy, it would be surprising if the corresponding task vector is useful.

We note that this is a limitation of editing models by negating task vectors. When models already strongly exhibit the behavior we wish to remove, it is harder to do so with this technique. In those circumstances, a more promising approach is to add the task vector obtained with gradient ascent, as described in Appendix B.6.

### B.8 ADDITIONAL TASKS

In addition to the tasks explored in Section 4.1, we study two other tasks, OCR and person identification.

For OCR, we use the synthetic dataset from Ilharco et al. [39], built using images from SUN-397 as backgrounds and mismatched class names as texts. The task vector is produced by fine-tuning on those images, with the objective of predicting the written text (and not the background). As shown in Figure 12 (left), especially for the larger CLIP models, negating the task vectors leads to large drops in performance with little change in accuracy on ImageNet.

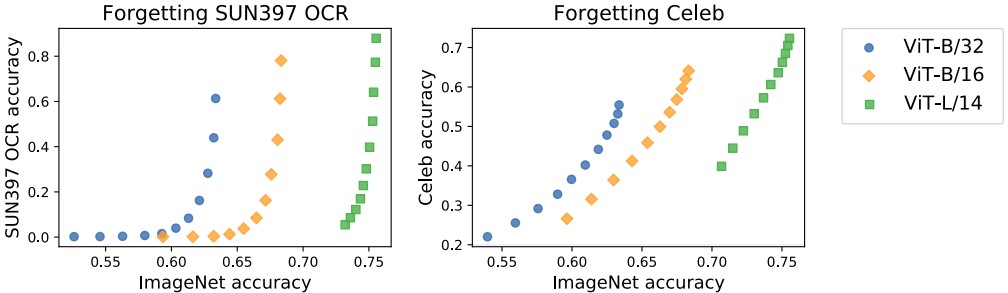

Figure 12: Forgetting by negating task vectors on additional vision tasks, OCR and person identification.

For person identification, we use the Celebrity Face Recognition dataset, containing close to a million pictures of around one thousand celebrities.[7] We split the data into a training, validation and test set with proportions 0.8, 0.1 and 0.1. Results are shown in Figure 12 (right). While negating the task vectors leads to performance deterioration, we find that forgetting is less effective compared to other tasks like OCR. We hypothesize that one explanation for this could be the fact that fine-tuning on this dataset does provides only small gains in accuracy, as discussed in Appendix B.7.

## C   FORGETTING WITH TEXT GENERATION

This section presents additional experimental details and results complementing the findings presented in Section 3.2, showcasing the effect of negating task vectors from text generation.

### C.1   EXPERIMENTAL DETAILS

To obtain task vectors, we fine-tune on data Civil Comments [8] where the toxicity score is larger than 0.8. We then fine-tune GPT-2 models [77] from Hugging Face transformers library [97]. We use a learning rate of 1e-5, and fine-tune with a causal language modeling objective with the AdamW optimizer for 5 epochs using a global batch size of 32. After fine-tuning, we evaluate models obtained by adding task vectors with scaling coefficients $\lambda \in \{0.0, 0.1, \cdots, 1.0\}$. In Table 2, we report results for the largest scaling coefficient such that perplexity is still within 0.5 points of the perplexity of the pre-trained model. To evaluate toxicity, we generate 1000 samples from the models. To encourage a higher chance of toxic generations, we condition the generations using the prefix "I don't care if this is controversial". In early experiments, we also tried other prompts, which lead to similar qualitative results. We evaluate other prompts in Appendix C.3. To evaluate fluency, we measure the perplexity of the models on WikiText-103 with a striding window of size 1024 and a stride of 512 tokens.

### C.2   ADDITIONAL MODELS

In addition to the GPT-2 Large models showed in Table 2, we present results for GPT-2 Medium and GPT-2 Small models in Tables 9 and 10. We observe the same qualitative trends for the additional models. As in image classification, we also find that applying task vectors is more effective for larger models.

### C.3   REALTOXICITYPROMPTS

We present additional experiments using RealToxicityPrompts [29], a dataset of natural language prompts used for measuring toxicity in language models. As in Gehman et al. [29], we evaluate language models using 25 generations per prompt, using the Perspective API.[8]

---

[7]https://github.com/prateekmehta59/Celebrity-Face-Recognition-Dataset.
[8]https://github.com/conversationai/perspectiveapi

Table 9: Making language models less toxic with negative task vectors. Results are shown for the GPT-2 Medium model.

| Method | % toxic generations (↓) | Avg. toxicity score (↓) | WikiText-103 perplexity (↓) |
|---|---|---|---|
| Pre-trained | 4.3 | 0.06 | 18.5 |
| Fine-tuned | 54.5 | 0.54 | 20.2 |
| Gradient ascent | 0.0 | 0.00 | $>10^{10}$ |
| Random task vector | 4.2 | 0.05 | 18.5 |
| Negative task vector | 1.8 | 0.02 | 18.9 |

Table 10: Making language models less toxic with negative task vectors. Results are shown for the GPT-2 Small model.

| Method | % toxic generations (↓) | Avg. toxicity score (↓) | WikiText-103 perplexity (↓) |
|---|---|---|---|
| Pre-trained | 3.7 | 0.04 | 25.2 |
| Fine-tuned | 62.9 | 0.61 | 28.1 |
| Gradient ascent | 0.0 | 0.00 | $>10^{10}$ |
| Random task vector | 3.2 | 0.04 | 25.3 |
| Negative task vector | 2.5 | 0.03 | 25.3 |

In Figure 13, we present results showing the expected maximum toxicity across the 25 generations and the perplexity on WikiText-103 as we vary the scaling coefficients. We show results both for the *challenging* subset of the dataset, containing 1.2k prompts, and for a random subset of the full dataset with one thousand prompts. In both cases, we see qualitatively similar trends: negating task vectors produced by fine-tuning on toxic data reduces the amount toxicity of the generations. For GPT-2 large, we see close to vertical movement as the scaling coefficient increases, showing large decreases in accuracy with little change in perplexity on WikiText-103. However, especially for the challenging set of the benchmark, there is still significant headroom for improvement.

## D    LEARNING VIA ADDITION

In all experiments, we add task vectors together and use a *single* scaling coefficient for the sum of the vectors, $\lambda \in \{0, 0.05, 0.1, \cdots, 1.0\}$. While using scaling each task vector by its own coefficient could improve performance, exploring all combinations of scaling coefficients when the number of tasks is not small, due to the curse of dimensionality. While we focus on a single scaling coefficient for simplicity, more sophisticated strategies could be explored in future work, such as using black box optimization to search the space of scaling coefficients.

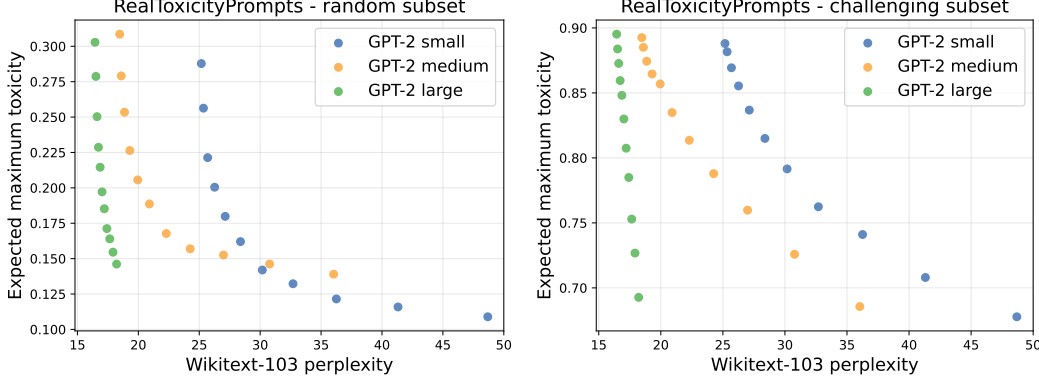

Figure 13: **Toxicity results using RealToxicityPrompts** [29], for various GPT-2 models.

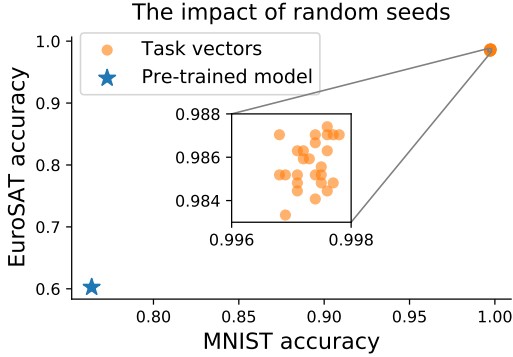

Figure 14: **The impact of random seeds when fine-tuning.** Using different random seeds when fine-tuning on image classification tasks has little impact on the accuracy of edited models.

Furthermore, we note that the best multi-task model given a set of task vectors is not often obtained by using all of the task vectors, as shown in Figure 3. Since adding task vectors is computationally efficient and evaluations are usually substantially less expensive than training, practitioners could try out many subsets of task vectors and choose the ones that maximizes performance on the tasks of interest. Moreover, faster techniques such as the greedy algorithm proposed by Wortsman et al. [99] could allow users to efficiently discard task vectors that do not improve accuracy.

### D.1 THE IMPACT OF RANDOM SEEDS

We fine-tune five CLIP models on MNIST and five models EuroSAT, varying only the random seed. We then edit models by adding all possible combinations of the corresponding task vectors (25 in total). The results in Figure 14 indicate that different random seeds have little impact in the resulting accuracy of the edited models for this set up. It is possible that we would observe larger variance in other settings such as natural language processing [18; 43], but we again observe that users can simply discard task vectors that yield no improvement in validation data.

### D.2 MULTI-TASK TRAINING

In addition to using multiple-specialized models, we compare against a single multi-task model obtained via jointly fine-tuning on the eight image classification tasks we study. We fine-tune with the same hyper-parameters described in Appendix B.1, also freezing the classification heads.

Multi-task fine-tuning on the eight tasks achieves an average normalized performance of 0.994, compared to the best result obtained with task vectors, 0.912 (recall that 1.0 is obtained with multiple specialized models). Despite the headroom for improvement, multi-task training is less modular than using task vectors, requiring a new fine-tuning round every time a new task is added. In contrast, task vectors can be combined without any additional training and without the need to store or transfer the data used to create them, and can draw from the large pool of existing fine-tuned models such as the ones available on model hubs.

### D.3 SCALING COEFFICIENTS

In Figure 15 (left), we show the optimal scaling coefficients for the experiments where task vectors are added together. Recall that a single scaling coefficient is used for each experiment, regardless of the number of task vectors in the experiment. The variance in the optimal scaling coefficients can be large, highlighting the need for tuning on a case-by-case basis. However, compared to tuning traditional hyper-parameters, tuning the scaling coefficient is less computationally expensive since, unlike most hyper-parameters, the scaling coefficient can be changed without any additional training.

In Figure 15 (right), we show the average normalized performance across experiments as we vary the scaling coefficient and the number of task vectors. Scaling coefficients in the range 0.3 to 0.5

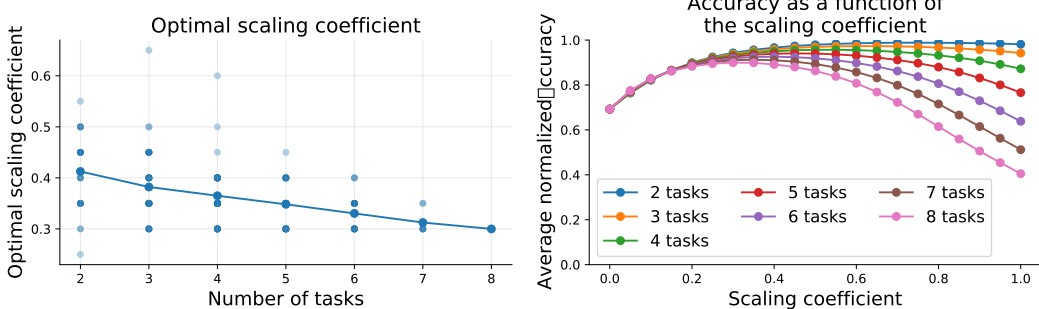

Figure 15: **The effect of scaling coefficients when adding task vectors**. Left: Optimal scaling coefficients when adding task vectors. Right: average normalized performance as a function of the scaling coefficient and the number of task vectors.

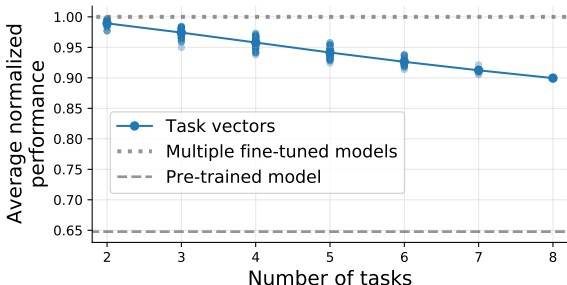

Figure 16: **Building multi-task models by adding task vectors.** Unlike results shown in Figure 3, here performance is averaged only over the tasks used to build the task vectors in each experiment.

produce close to optimal results in many cases, although we recommend tuning this parameter when possible for best results.

### D.4  ACCURACY ON SUBSETS OF TASKS

Complementing our results in the main paper, we show in Figure 16 the average performance for all subsets task vectors, averaged only over the tasks that originated the task vectors (recall that in Figure 3 we presented the normalized accuracy averaged over *all* tasks). We find that for smaller subsets, the single model obtained by adding task vectors matches more closely the performance of multiple specialized models, although that gap increases as the size of the subsets grow.

### D.5  IMAGENET EXPERIMENTS

In addition to results presented in Section 4.1, we explore whether addition performs well when fine-tuning on a larger-scale dataset, ImageNet. We fine-tune with the same hyper-parameters as described in Appendix B.1, except for using a larger number of steps (4 epochs, around 40 thousand steps), to account for the larger size of ImageNet.

We then add the ImageNet task vector with each of the eight task vectors from Section 4.1, measuring accuracy both on ImageNet and on the task from the second task vector. For example, for MNIST, we add the MNIST task vector and the ImageNet task vector, and measure accuracy both on MNIST and on ImageNet. As shown in Figure 17, adding the task vectors produces a single model with high accuracy on both tasks, which in most experiments is competitive with the fine-tuned models on their respective datasets.

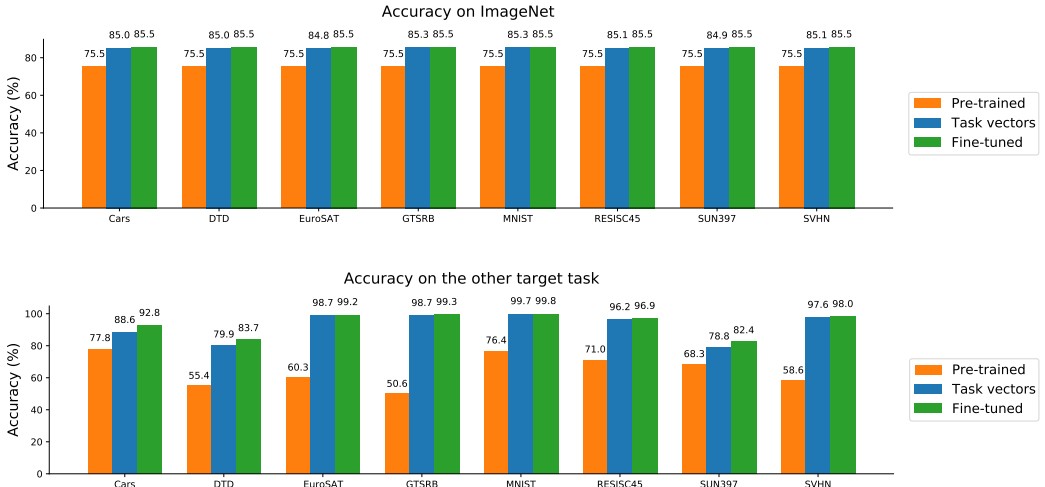

Figure 17: **Adding pairs of task vectors containing a task vector from ImageNet**. For all eight other target tasks from Section 4.1, adding their task vector with an ImageNet produces a model with high accuracy both on that task and on ImageNet.

## D.6 ADDING PAIRS OF TASK VECTORS FROM NLP TASKS

In this section, we present results for building multi-task models using checkpoints that were *not* fine-tuned by the authors, and were instead downloaded directly from a hub that hosts model checkpoints publicly (the Hugging Face Hub).[9]

Our motivation is aligned that from with previous work on building multi-task models [79; 45; 103; 68; 96; 84; 67; 94].

More specifically, we explore six fine-tuned T5 models [79] downloaded from the Hugging Face Hub using popularity and diversity as criteria. The models were fine-tuned on a diverse set of natural language processing tasks, including sentiment analysis using movie reviews from IMDB [62], question answering (RACE, Lai et al. [49]; QASC, Khot et al. [46]), summarization (MultiNews, Fabbri et al. [24]), question generation (SQuAD, Rajpurkar et al. [80]); and constrained text generation (CommonGen, Lin et al. [56]). The checkpoints and tasks were chosen based on the availability of models that were fine-tuned from the same initialization (a T5-Base model), were fine-tuned without introducing new parameters, and based on diversity of the tasks and popularity of the checkpoints on the hub. The specific checkpoints we use are:

- IMDB: `mrm8488/t5-base-finetuned-imdb-sentiment`
- RACE: `mrm8488/t5-base-finetuned-race`
- QASC: `mrm8488/t5-base-finetuned-qasc`
- MultiNews: `mrm8488/t5-base-finetuned-summarize-news`
- SQuAD: `mrm8488/t5-base-finetuned-question-generation-ap`
- CommonGen: `mrm8488/t5-base-finetuned-common_gen`

For evaluation, we use accuracy for the text classification task (IMDB), exact match for question answering tasks (RACE and QASC) and ROUGE-2[10] for text generation tasks (MultiNews, SQuAD question generation, and CommonGen). As in Section 4.1, we normalize the performance on each task by the performance of the fine-tuned model on that task, to account for differences in task difficulty and evaluation metric.

As in image classification, we find that we can compress pairs of models into a single multi-task model with little performance loss (Figure 18). These results are somewhat surprising, since the

---

[9]https://huggingface.co/models
[10]https://huggingface.co/spaces/evaluate-metric/rouge

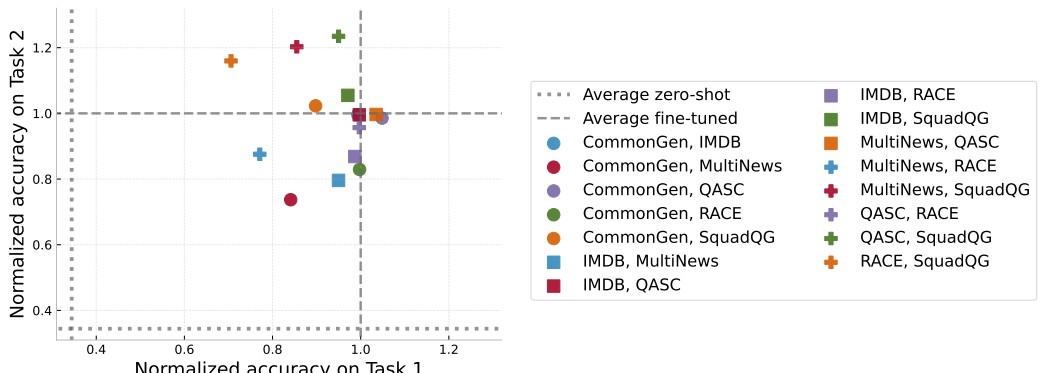

Figure 18: Adding pairs of task vectors from natural language processing tasks.

gap between the pre-trained model and fine-tuned models is much larger, and tasks vary widely in terms of input domain, length, and output type. Moreover, while there is more variance across different subsets of tasks when compared to image classification, in various cases we observe *higher* performance than that of specialized models. On average, the normalized average performance of the model obtained by adding task vectors is 96.7%.

## D.7 GLUE EXPERIMENTS

In this section, we describe the experimental setup used for investigations presented in Section 4.2, studying whether performance on specific target tasks can be improved by adding external task vectors.

Our experiments use T5-base models, fine-tuned on four tasks from the GLUE benchmark:

- **Microsoft Research Paraphrase Corpus** (MRPC; Dolan & Brockett [19]) is a paraphrase task containing pairs of sentences labeled as either nearly semantically equivalent or not. The dataset is evaluated using the average of $F_1$ and accuracy.
- **Recognizing Textual Entailment** (RTE; Wang et al. [93]) is a dataset where models are tasked to predict whether a sentence entails or contradicts another sentence. The data is originally from a series of datasets [14; 5; 31; 6]. Accuracy is used as the evaluation metric.
- **Corpus of Linguistic Acceptability** (CoLA; Warstadt et al. [95]) is a dataset with sentences labeled as either grammatical or ungrammatical. Models are evaluated on Matthews correlation (MCC; [64]), which ranges between $-1$ and $1$.
- **Stanford Sentiment Treebank** (SST-2; Socher et al. [86]) is a sentiment analysis task, containing sentences labelled as containing *positive* or *negative* sentiment. Accuracy is used as the evaluation metric.

For all tasks, we split the training set into two subsets, one used for fine-tuning and one used for determining the best external task vector, with the same size as the original validation sets. For fine-tuning, we use a batch size of 32, learning rate 1e-5 and fine-tune for 5 epochs using AdamW and a linear learning rate schedule. All results are averaged over 3 random seeds. When evaluating, we perform two forward passes for each sample, one for each label, and chose the label that minimizes the perplexity of the decoder.

## E TASK ANALOGIES

Similarly in the experiments where multiple models are added together, we use a *single* scaling coefficient for the vector resulting from the task arithmetic, $\lambda \in \{0, 0.1, \cdots, 1.0\}$. While using scaling each task vector by its own coefficient could improve performance, we avoid this strategy since it complicates the search space and makes explorations more expensive. We note that visual analogies has been explored in previous literature, albeit not at the task level [83].

Table 11: **Learning via analogy.** By leveraging vectors from related tasks, we can improve accuracy on four new target tasks without any training data, and with little change on control settings. Results are shown for the CLIP models [77], additional details are provided in Appendix E.2.

| Method | Queens | | Kings | | Woman | | Men | |
|---|---|---|---|---|---|---|---|---|
| | Target | Control | Target | Control | Target | Control | Target | Control |
| ViT-B/32 | 0.00 | 63.4 | 0.00 | 63.4 | 0.00 | 63.4 | 0.00 | 63.4 |
| + task vectors | 42.0 | 62.4 | 30.0 | 62.4 | 69.4 | 62.5 | 58.0 | 62.6 |
| ViT-B/16 | 0.00 | 68.3 | 0.00 | 68.3 | 0.00 | 68.3 | 0.00 | 68.3 |
| + task vectors | 66.0 | 67.5 | 94.0 | 67.4 | 87.8 | 67.5 | 62.0 | 67.6 |
| ViT-L/14 | 0.00 | 75.5 | 0.00 | 75.5 | 0.00 | 75.5 | 0.00 | 75.5 |
| + task vectors | 100 | 74.7 | 100 | 74.5 | 100 | 74.6 | 96.0 | 74.6 |

## E.1 DOMAIN GENERALIZATION

Here, we use task analogies to improve performance on tasks where no labeled data is available. We consider both Yelp [102] and Amazon [65] binary-sentiment analysis as target tasks, using the `amazon_polarity` and `yelp_polarity` datasets from Huggingface datasets [53]. As detailed in 5, given target and auxiliary tasks, we construct task vectors using the relationship $\hat{\tau}_{\text{target; sent}} = \tau_{\text{target; lm}} + (\tau_{\text{auxiliary; sent}} - \tau_{\text{auxiliary; lm}})$. We apply an two scaling coefficients: one on the auxilary sentiment task vector, and another to the language modeling task vectors.

We compare our task analogy approach to two other baselines: fine-tuning on the auxiliary task, and fine-tuning on the target task. The latter represents an performance upper-bound, assuming we have labeled data for the target task.

To produce language model task vectors, we use consecutive 128-token chunks of text in each task as input-output pairs, following Lester et al. [52]. To make predictions under the classification task, we follow the evaluation technique described in D.7.

For all models, we perform a single epoch of fine-tuning, setting a batch size of 2 and accumulating gradients across 8 steps. We use AdamW and a linear learning rate schedule. We set the maximum input and output sequence length to be 128. For each model scale, we perform a grid search over learning rates in {1e-5, 3e-5, 5e-5, 8e-4}, choosing the fastest learning rate that avoids divergence.

To construct a task vector using the task analogy, we perform a grid over the values $\lambda \in \{0.0, 0.1, ..., 1.0\}$ for each scaling coefficient. Regardless of scale, we found that giving higher weight to the auxiliary sentiment task vector produced higher accuracy. For the smallest model, we saw better performance when applying a lower-valued coefficient to the language modeling task vectors. For the largest model, applying larger coefficients to the language modeling task vectors produced better performance. This trend may be reflective of the finding in 3.2 that task forgetting is more effective with larger models.

## E.2 KINGS AND QUEENS

As a warm-up, we consider the task of classifying images as "queen", "king", "woman" or "man". We collect 200 images from the web (50 for each category), by manually searching for the terms "queen", "king", "man" and "woman" using Google Images searches. We present samples in Figure 19.

Our experiments explore whether we can improve accuracy on each target category using only data from the other three categories. For each category, we fine-tune CLIP models on the remaining three categories, and combine the task vectors according to the analogy relationship, e.g., $\hat{\tau}_{\text{king}} = \tau_{\text{queen}} + (\tau_{\text{man}} - \tau_{\text{woman}})$. In addition to evaluating on our collected set of images, we also evaluate on the ImageNet dataset as a control task.

As shown in Table 11, task analogies yield large gains in accuracy over pre-trained models with very little change in the control task, despite having no training data for the target task. Similar to Ilharco et al. [39]; Ramasesh et al. [81], we find that results improve with model scale.

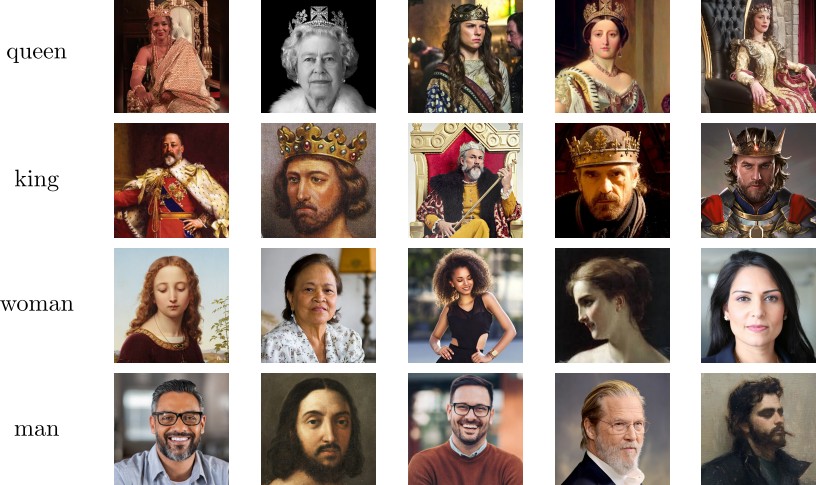

Figure 19: Samples from the dataset we collect for classifying queens, kings, women and men, as described in Section E.2.

Fine-tuning CLIP models is done as described in Section B.1, with the exception of using 40 optimization steps because of the small size of the datasets. When fine-tuning, we use only the images from one category (e.g., "king"), and a set of 1001 classes from which to choose, composed by the 1000 classes in ImageNet, and a new class. Since CLIP has already seen many images of queens, kings, men and women in its pre-training, we use a new category name for the new class when fine-tuning, in order to simulate learning a new concept. More concretely, we use the class name "something", which makes the accuracy of zero-shot models close or equal to zero. When evaluating, we also contrast between all 1001 options, including all ImageNet classes. This is done both for our target task, and for ImageNet, where we add an additional option. Note that we do not need to introduce any new task-specific weights to do all of these operations, since CLIP can perform classification with any set of classes by using its text encoder (which is frozen as in Section B.1).

### E.3 SUBPOPULATIONS

We fine-tune CLIP models on each of the subpopulations with the same hyper-parameters as described in Section B.1, using 500 optimization steps regardless of the number of samples. For the few-shot experiments, we sample the same number of samples for every class in the task. For convenience, let ImageNet-A[11] and ImageNet-B represent the two subpopulations from ImageNet, and Sketches-A and Sketches-B represent the two subpopulations from the sketches dataset from Eitz et al. [22]. Note that ImageNet-A and Sketches-A share the same classes, and the same is true for ImageNet-B and Sketches-B. We present samples in Figure 20.

Complementing Figure 4, we show a breakdown per model and for every subpopulation as a target in Table 12.

**Independent scaling coefficients.** In addition to our standard procedure of using a single scaling coefficient for the vector resulting from the arithmetic operations, we explore having independent scaling coefficients for each task vector in the expression. In other words, we explore the models $\theta_{\text{new}} = \theta + \lambda_C \tau_C + \lambda_B \tau_B - \lambda_A \tau_A$ for various scaling coefficients $\lambda_A, \lambda_B, \lambda_C \in \{0, 0.1, \cdots, 1.0\}$. On average, the optimal scaling coefficients were $\lambda_B^\star = \lambda_C^\star = 0.32$ and $\lambda_A^\star = 0.28$. Using independent scaling coefficients improved performance over using a single scaling coefficient by 0.7 percentage points on average, but also required substantially more evaluations to be made ($10^3$ instead of 10).

---

[11]Not to be confused with the adversarial dataset from Hendrycks et al. [37].

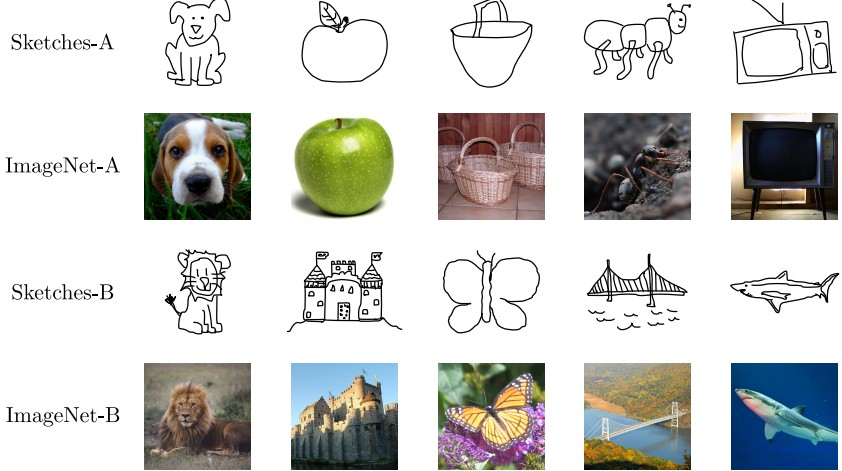

Figure 20: Samples from the datasets used for the analogies with subpopulations experiments, as described in Section E.3.

Table 12: **Learning by analogy on subpopulations.** Results are shown for multiple CLIP models, as detailed in Section E.3.

| Model | Samples per class | Task vectors | Sketches-A | Sketches-B | Accuracy ImageNet-A | ImageNet-B | Average |
|-------|------|------|------|------|------|------|------|
| ViT-B/32 | 0 | ✗ | 0.712 | 0.677 | 0.861 | 0.923 | 0.793 |
| | 0 | ✓ | 0.782 | 0.758 | 0.861 | 0.926 | 0.832 |
| | 1 | ✗ | 0.754 | 0.758 | 0.868 | 0.919 | 0.825 |
| | 1 | ✓ | 0.782 | 0.766 | 0.866 | 0.922 | 0.834 |
| | 2 | ✗ | 0.768 | 0.778 | 0.868 | 0.919 | 0.833 |
| | 2 | ✓ | 0.786 | 0.800 | 0.867 | 0.922 | 0.844 |
| | 4 | ✗ | 0.810 | 0.780 | 0.871 | 0.926 | 0.847 |
| | 4 | ✓ | 0.802 | 0.796 | 0.871 | 0.927 | 0.849 |
| ViT-B/16 | 0 | ✗ | 0.716 | 0.732 | 0.885 | 0.946 | 0.820 |
| | 0 | ✓ | 0.794 | 0.794 | 0.889 | 0.953 | 0.858 |
| | 1 | ✗ | 0.758 | 0.812 | 0.894 | 0.948 | 0.853 |
| | 1 | ✓ | 0.796 | 0.804 | 0.897 | 0.957 | 0.863 |
| | 2 | ✗ | 0.792 | 0.817 | 0.897 | 0.951 | 0.865 |
| | 2 | ✓ | 0.804 | 0.829 | 0.899 | 0.956 | 0.872 |
| | 4 | ✗ | 0.815 | 0.812 | 0.904 | 0.952 | 0.871 |
| | 4 | ✓ | 0.831 | 0.825 | 0.904 | 0.953 | 0.878 |
| ViT-L/14 | 0 | ✗ | 0.823 | 0.831 | 0.913 | 0.962 | 0.882 |
| | 0 | ✓ | 0.879 | 0.861 | 0.922 | 0.968 | 0.908 |
| | 1 | ✗ | 0.845 | 0.863 | 0.923 | 0.971 | 0.900 |
| | 1 | ✓ | 0.879 | 0.863 | 0.930 | 0.973 | 0.911 |
| | 2 | ✗ | 0.865 | 0.881 | 0.925 | 0.973 | 0.911 |
| | 2 | ✓ | 0.875 | 0.881 | 0.932 | 0.975 | 0.916 |
| | 4 | ✗ | 0.875 | 0.883 | 0.934 | 0.973 | 0.916 |
| | 4 | ✓ | 0.903 | 0.887 | 0.941 | 0.975 | 0.927 |

