# OpenReview forum: "Editing models with task arithmetic"
_ICLR.cc/2023/Conference — ICLR 2023 poster_

### Official Review · Reviewer_t87y · 2022-10-24

**Confidence:** 3
**Clarity, Quality, Novelty And Reproducibility:** Presentation is clear, and idea seems…
**Correctness:** 3
**Technical Novelty And Significance:** 3
**Empirical Novelty And Significance:** Not applicable
**Recommendation:** 5

**Strength And Weaknesses:**

## Strength
- The idea seems to be novel and interesting. If we can add multiple task vectors without any interference, it would be very impactful result in terms of multi-task learning since task interference is one of the most important problem to be tackled in multi-task learning community.

-  The authors have performed various experiments (image classification and natural language processing tasks) to verify the proposed method.

## Weakness
- The paper lacks of analysis about why the proposed methods work. For example, the authors show that multiple task vectors works better than fine-tuning the pre-trained model with multi-task learning objective. But they do not explain why the proposed method show better performance in terms of optimization.

- Figure 2 does not seem to be informative enough. In section 4.1, they claim that the proposed method achieves higher accuracy than the multiple fine-tuned models, but Figure 2 does not show anything about the multiple fine-tuned models. Comparing the proposed method and pre-trained model is not informative since that comparison is too trivial and not surprising.

- I think authors should compare the proposed method to stronger baselines to rigorously verify effectiveness of their method. For example, for the experiments with the addition of task vector, I highly recommend to add some multi-task learning baselines such as [1, 2]. Moreover, since the authors tune $\lambda$ with validation set, we can take a set of snapshots of the model during finetuning and select one of the snapshot that achieves the best validation performance for fair comparison.


[1] Yu, Tianhe, et al. "Gradient surgery for multi-task learning." Advances in Neural Information Processing Systems 33 (2020): 5824-5836.

[2] Navon, Aviv, et al. "Multi-task learning as a bargaining game."ICML 2022.

**Summary Of The Paper:**

This paper proposes task vectors, which are subtraction of pre-trained weights from the weights fine-tuned on downstream tasks,
and perform add or subtract the task vector to the pre-trained weight to steer pre-trained models. With the task vector, we can remove undesirable behavior  of pre-trained model (e.g. generation of toxic sentences) by negating the task vector and adding it to the pre-trained model. Moreover, we can perform multi-task learning, by obtaining  independently task vector for each task and summing all the task vectors to the pre-trained weight. Lastly, similar to word2vec, we can perform analogy with the task vector.

**Summary Of The Review:**

I am inclined to reject since the experiments are not thorough enough to verify the effectiveness of the proposed method and the paper lacks of analysis why the proposed method work. However, if the authors can address my concerns, I will be happy to raise my score.

---

> ### Author Response · Authors · 2022-11-15
> **Response to Reviewer t87y**
>
> Thank you for your thoughtful comments and suggestions, and thank you for engaging with us to make our work stronger. We address individual questions and concerns below.  We would greatly appreciate it if you took the new experiments and clarifications below into account during discussions, and if there is anything else we can do to further improve our work, please let us know.
>
> ***
>
> **Why does our method work?** We substantially expanded our discussion on why our method works following this review. More specifically, in the new **Appendix E**, we examine the connection between weight averaging and the well-established technique of ensembling, which is widely used to combine models. This connection is discussed in depth in Wortsman et al., 2022ab [7, 8], and we revisit it in the context of adding task vectors. First, recall that all arithmetic operations on top of task vectors are linear combinations---or weighted averages---of model weights. As shown by Wortsman et al., 2022 [7, 8], in certain regimes, **linear combinations of neural network weights approximate ensembling their outputs**. This approximation holds whenever the objective landscape can be approximated by a linear expansion [12], which has been shown to become more accurate for models that were already trained for many steps [6], such as when initializing from a pre-trained model.
>
> We empirically validate this connection in the context of adding two task vectors. Note that the model resulting from adding two task vectors with a scaling coefficient of 0.5 is equivalent to an unweighted average of the weights of the fine-tuned models. We then investigate whether accuracy of the model obtained using the task vectors correlates with accuracy of ensembling the fine-tuned models, as predicted by theory. As shown in **Figure 18 in Appendix E**, we indeed observe that **the accuracy of the model produced by adding task vectors closely follows the accuracy of the ensemble** of the corresponding fine-tuned models, and the two quantities are strongly correlated, with a Pearson correlation of 0.99.
>
> Finally, we note that several authors have also empirically observed high accuracy when averaging two neural networks that share part of their optimization trajectory (such as the same pre-trained initialization) [3-11], as we discuss in Appendix E. Upon acceptance, we would bring some of this important discussion into the main paper.
>
> ***
>
> **Figure 2.**  Recall that in Figure 2 we display performance *normalized by that of fine-tuned models*. This means that, in this plot, y=1.0 represents the performance of using multiple fine-tuned models. We thank the reviewer for the comment, and note that Figure 2 has been updated to highlight this fact.
>
>
> ***
>
> **Stronger multi-task baselines.** When building systems capable of handling multiple tasks, the optimal method depends on training and deployment constraints. When computational and memory requirements at deployment time are not a concern, training and using *multiple specialized models* is the best option, as it shows the highest accuracy. *Multi-task methods such as [1,2]* are better suited for cheaper deployment since only one model needs to be served, but require more expensive training compared to task vectors, since the modifications to fine-tuning come with an overhead. For these methods, a new model needs to be trained for every unique set of tasks, and data from all tasks is required during fine-tuning. Finally, *task vectors* are computationally cheap to produce and deploy, can re-use already trained models (as we discussed in Section 4.2), and are ideally suited for scenarios such as federated learning where data can’t be shared due to privacy or other constraints.
>
> Following this review, **we ran additional experiments** with two recent multi-task learning methods, PCGrad [1] and Nash-MTL [2]. Since these methods require an entire training run for each subset of tasks, we ran experiments in only two settings: i) for the pair of tasks MNIST and EuroSAT, and ii) in the most challenging setting where all eight tasks are used, and where performance with task vectors is lowest. For MNIST and EuroSAT, all methods substantially improved the accuracy of the zero-shot model (68.3% on average), with an average accuracy of 99.2% for task vectors, 99.4% for both PCGrad and Nash-MTL, and 99.5% when using two specialized models. In the more challenging scenario where all tasks are used, PCGrad and Nash-MTL also performed better than adding task vectors, but worse than using specialized models (95.1% for PCGrad and 96.2% for Nash-MTL, compared to the 90.0% average normalized performance for task vectors, and 100% when using multiple specialized models). We show this new result in our updated Figure 2.  Overall, we find a **trade-off between different methods**, and **recommend different approaches for different circumstances**, depending on the data, compute and memory constraints for training and deployment.

---

> > ### Author Response · Authors · 2022-11-15
> > **References**
> >
> > **References**
> >
> > [1] Yu, Tianhe, et al. Gradient surgery for multi-task learning. NeurIPS 2020.
> >
> > [2] Navon, Aviv, et al. Multi-task learning as a bargaining game. ICML 2022.
> >
> > [3] Frankle, Jonathan, et al. Linear mode connectivity and the lottery ticket hypothesis. ICML, 2020.
> >
> > [4] Izmailov, Pavel, et al. Averaging weights leads to wider optima and better generalization. In Conference on Uncertainty. UAI 2018.
> >
> > [5] Neyshabur, Behnam, et al. What is being transferred in transfer learning? NeurIPS, 2020.
> >
> > [6] Fort, Stanislav, et al.. Deep learning versus kernel learning: an empirical study of loss landscape geometry and the time evolution of the neural tangent kernel. NeurIPS, 2020.
> >
> > [7] Wortsman, Mitchell, et al. Robust fine-tuning of zero-shot models. CVPR, 2022.
> >
> > [8] Wortsman, Mitchell, et al. Model soups: averaging weights of multiple fine-tuned models improves accuracy without increasing inference time. ICML, 2022.
> >
> > [9] Choshen, Leshem, et al. Fusing finetuned models for better pretraining, 2022.
> >
> > [10] Lucas, James, et al. Analyzing monotonic linear interpolation in neural network loss landscapes. 2021.
> >
> > [11] Ilharco, Gabriel, et al. Patching open-vocabulary models by interpolating weights. NeurIPS, 2022.
> >
> > [12] Jacot, Arthur, et al. Neural tangent kernel: Convergence and generalization in neural networks. NeurIPS, 2018.

---

### Official Review · Reviewer_9yQa · 2022-10-24

**Confidence:** 3
**Correctness:** 3
**Technical Novelty And Significance:** 3
**Empirical Novelty And Significance:** 3
**Recommendation:** 6

**Clarity, Quality, Novelty And Reproducibility:**

In my opinion, arithmetic operations with task vectors are novel and the results are also interesting.  Code repo is  provided in the main paper for the reproducibility. However, I can not find the code at https://github.com/redacted and i am not sure this repo breaks the Blind Submission but i can find the name “Steven Tobin” and other information.


**Strength And Weaknesses:**

Strength

The definition of a task vector might be a matter of course because the pre-trained model can be considered as an updated model with meaningful directions. However, arithmetic operations with these task vectors are very interesting. For example,
Compared to traditional multitask learning which is updated from the total loss ( Loss A + Loss B) Learning via addition (Task vector A + Task vector B) can improve the pre-trained model on the tasks, too



Weaknesses

From the definition of a task vector, arithmetic operations can be applied to the same structure model because of the dimension of the model.
In the image classification task, more realistic dataset such as ImageNet and the deep model such as ResNet-50 are needed for verifying the effectiveness of the task vector and the possibility for the realistic dataset.


**Summary Of The Paper:**

In this work, they propose a new paradigm based on the task vectors.
This task vector implies a direction of the pre-trained model. Task vectors are defined as a subtraction between a pre-trained model and fine-tuned model from a same pre-trained model with a specific task. This vectors can be used with arithmetic operations such as negation and addition.


**Summary Of The Review:**

In this work, they propose a new paradigm named task vectors. For various vision and NLP models, adding multiple specialized task vectors results in a single model that performs well on all target tasks. Arithmetic operations are very simple but effective.

---

> ### Author Response · Authors · 2022-11-15
> **Response to Reviewer 9yQa**
>
> Thank you for your valuable comments and suggestions, and for highlighting that our method is novel and our results interesting. We apologize for the misunderstanding regarding the link in our paper. This link was provided only as a placeholder, and bears no connection with this project or any of the authors. Please find our responses to your other comments below, and please let us know if there is anything else we can do to further strengthen our submission.
>
> ***
>
> **Realistic datasets.** We added new experiments adding task vectors from image classification tasks. More precisely, we explored whether addition performs well when fine-tuning on a larger-scale dataset, **ImageNet**. We fine-tune with the same hyper-parameters as described in Appendix A.1, except for using a larger number of steps (4 epochs), to account for the larger size of ImageNet. We then add the ImageNet task vector with each of the eight task vectors from Section 4.1 (Cars, DTD, EuroSAT, MNIST, GTSRB, RESISC45, SUN397 and SVHN), measuring accuracy both on ImageNet and on the task corresponding to the second task vector. For example, for MNIST, we add the MNIST task vector and the ImageNet task vector, and measure accuracy both on MNIST and on ImageNet. Our new results are presented in **Figure 15 (Appendix C.3)**. We find that **adding task vectors is effective in all experiments**, producing a single model with high accuracy on both tasks. In all experiments, the model obtained by adding task vectors is competitive with the fine-tuned models on their respective datasets (on average, within 1 percentage point in accuracy).
>
> We also highlight that our image classification experiments use large-scale vision transformers (ViTs) as backbones, which are deep models, and presently the backbone of state-of-the-art models in many image classification tasks including ImageNet.
>
> ***
>
> **Changes in model architecture.** As noted by the reviewer and by the authors in Section 6, task vectors are restricted to models with the same architecture, since they depend on element-wise operations on model weights. We note that, despite this limitation, some architectures are very popular, and encompass the majority of models fine-tuned by the community, especially in NLP. While outside of the scope of this work, one exciting way to bridge task vectors from different architectures would be to perform distillation with the fine-tuned models [1].
>
> ***
>
> **References**
>
> [1] Hinton, Geoffrey, Oriol Vinyals, and Jeff Dean. "Distilling the knowledge in a neural network." arXiv preprint arXiv:1503.02531 2.7 (2015).

---

### Official Review · Reviewer_gv8P · 2022-10-24

**Confidence:** 4
**Clarity, Quality, Novelty And Reproducibility:** The method is very simple and clear a…
**Correctness:** 3
**Technical Novelty And Significance:** 3
**Empirical Novelty And Significance:** 3
**Recommendation:** 5

**Strength And Weaknesses:**

Strength.
+ This paper proposes a very simple and general method which I can imagine has wide applications with different use cases.
+ The method is supported with a wide range of experiments across image classification language generation.
+ The visualisation of cosine similarity shows why the proposed method works and has very little interference.

Limitations.
+ The provided link points to a private github repo which might breaks the anonymity requirement for ICLR.
+ The paper mentions the scaling term lambda, which is a learnable parameter based on validation performance. However, I did not see any additional experiments or explanation in the experiment section?
+ There are also no explanations on the gradient ascent and random vector baselines. Is it doing simply gradient descent for the fine-tuning baseline but reverse the gradient signs? But if all task weights are orthogonal to each other why gradient ascent would affect control task performance?


**Summary Of The Paper:**

This paper presents a very simple technique to edit pre-trained model using arithmetic on network weights.  From which, the paper evaluates three ways to do athematic: i) negation: to reduce the performance for a particular dataset, ii) addition: add the performance for a dataset which results a multi-task model, and iii) analogy: to improve performance for a dataset with fewer training data by adding and subtracting two related datasets via analogies. A similar concept like king = queen – women + men, as shown in semantic embedding in a language model. The paper verifies the proposed approach in both vision and language domain.

**Summary Of The Review:**

The paper proposes a simple method to modify a pre-trained model by weight arithmetic. The idea is simple and interesting and has the potential to be a good and general solution to a lot of applications. But the paper itself is very rushed to make, with some experiment details and settings missing. And I believe the authors break the anonymity requirement with a personal GitHub link.

---

> ### Author Response · Authors · 2022-11-15
> **Response to Reviewer gv8P**
>
> Thank you for your very useful suggestions, and for engaging with us to improve our work. Please see below the response to your concerns, which include a number of new analyses, and please let us know if there is anything else we can do to further improve our paper.
>
> ***
>
> **Github link:** the github link in our paper (github.com/redacted) is a placeholder, and is not related in any capacity with this paper or any of the authors. As such, we do not believe it violates the anonymity requirement for this conference. We used the word “redacted” on the link as a placeholder, which means to “censor or obscure (part of a text) for legal or security purposes”. We sincerely apologize for the confusion.
>
> ***
>
> **Scaling coefficient:** Following this review, **we added new analysis** on the impact of the scaling coefficient when adding task vectors. More specifically, **Figure 13 in Appendix C** shows the optimal scaling coefficient for the experiments in the main paper, and **Figure 14 in Appendix C** shows how performance varies as a function of the scaling coefficient. Recall that the scaling term lambda is a single scalar used as a hyper-parameter in each of our experiments---for simplicity and efficiency, we do not use multiple scaling coefficients when adding more than one task vector. We observe some variance in the optimal scaling coefficients, with smaller coefficients performing better when larger numbers of tasks vectors are used. **In most cases, scaling coefficients on the range 0.3-0.5 perform well.** However, when possible, *we recommend tuning the scaling coefficient*, and note that this can be done much more efficiently than hyper-parameter searches. This is because, unlike most hyper-parameters, the scaling coefficient can be changed without any additional training, as it only controls how already trained models are combined.
>
> ***
>
> **Gradient ascent and random vector baselines:** Building on this review, we have included a **new and more detailed explanation of the baselines in Appendix A.2**. In short, gradient ascent is performed by fine-tuning with a negative cross-entropy loss, and random vectors are drawn layer-wise from a gaussian distribution, then scaled to match the magnitude of the corresponding layer of the task vector. More detailed explanations are found in the revised manuscript. Finally, we address the question of why gradient ascent can affect performance on the control task. We believe this is due to catastrophic forgetting of previously learned behavior [1], i.e. fine-tuning a pre-trained model (with gradient ascent or otherwards) can  overspecialize the model towards fitting the finetuning data, to the detriment of other skills. Indeed, this is what we see in practice: the model gets better at its optimization objective, but performance on the control task deteriorates. Finally, we want to make it clear that we only ever use control task data for evaluation, i.e. we don’t compute task vectors for the control tasks, nor do we measure orthogonality with respect to them.
>
> ***
>
> **References:**
>
> [1] Michael McCloskey and Neal J Cohen. Catastrophic interference in connectionist networks: The sequential learning problem. In Psychology of Learning and Motivation. Elsevier, 1989.

---

### Author Response · Authors · 2022-11-15
**Response to all reviewers**

We kindly thank all the reviewers for their time and thoughtful feedback. We are grateful that reviewers have pointed out that our method is novel (9yQa, t87y), interesting (gv8P, 9yQa, t87y), simple (gv8P, 9yQa, t87y), effective (9yQa) and impactful (t87y). We also thank reviewers for highlighting that our work covers a wide range of experiments (9yQa, t87y) and shows why our method works (gv8P).

Building on the reviews, we ran several new experiments and included additional discussions and analyses that can be found in our revised paper. The changes include Figures 2, 6, 13, 14, 15, 18, and Appendices A.2, C.2, C.3, and E. We hope these new experiments and analyses, in addition to the clarifications detailed in the individual responses, address the remaining concerns raised by the reviewers.

---

### Author Response · Authors · 2022-12-06
**Follow up**

Dear Reviewers,

We’d like to reach out again to check if there were any additional questions or concerns about our rebuttal that we can address before the  discussion period ends. Thanks again for taking the time to read our work and to provide helpful feedback, and please let us know if there is anything we can do to further improve our paper.

Paper Authors

---

### Decision · Program_Chairs · 2023-01-20

**Decision:**

Accept: poster

**Justification For Why Not Higher Score:**

Because of the violation of anonymity. If it is not an issue, I would strongly recommend Accept (poster).

**Justification For Why Not Lower Score:**

N/A

**Metareview: Summary, Strengths And Weaknesses:**

The paper proposes a novel concept, task vectors. Task vectors are subtraction of pre-trained weights from the weights fine-tuned on downstream tasks. The authors show that applying some arithmetic operations such as addition and negation on the task vectors has the effect of improving the performance or mitigating some biases. The claim is supported by extensive experiment results.

Strength. The proposed idea is very novel and interesting. It helps us understand better the relationship between pretrained weights and finetuned weights. I would expect more follow-ups in this direction. The experiments are also thorough. During rebuttal, the authors added more experiments and analysis asked by the reviewers.

Weakness. Why it should work in that sense is not clear although the authors tried to add some explanations in the rebuttal. But, I think this is fine. A paper doesn't need to solve all mysteries within a single paper. I think the current demonstration is already interesting enough. There was a concern about anonymity as the authors share a github link "https://github.com/redacted" and that link is currently connected to a github account of Steven Tobin.